# SIRT5-mediated desuccinylation of PPA2 enhances HIF-1alpha-dependent adaptation to hypoxic stress and colorectal cancer metastasis

Xiang Zhang [1,9], Yuqin Di [1,2,9], Youpeng Wang [3,9], Jiale Qin [1], Lvlan Ye [4], Xiangqiong Wen[5], Zunfu Ke [2,6 ✉], Ziyang Wang [7 ✉] & Weiling He [8 ✉]

## Abstract

Metastasis is the primary cause of death in patients with colorectal cancer (CRC). Hypoxia is a hallmark of solid tumors that promotes cellular metabolic adaptation and dissemination. However, the mechanisms linking hypoxia-regulated metabolic adaptation to CRC metastasis remain unclear. Here, we found that inorganic pyrophosphatase 2 (PPA2) suppresses metastatic progression of CRC via its phosphatase function. PPA2 expression levels are reduced in CRC specimen and correlate with enhanced response to hypoxia by promoting hypoxia-inducible factor-1 (HIF-1) signaling to promote CRC cell glycolysis and dissemination. Mechanistically, PPA2 decreases HIF-1alpha stability through non-canonical ubiquitin-mediated proteasomal degradation via recruitment of E3 ligase NEDD4. Furthermore, PPA2 directly dephosphorylates NEDD4 at threonine 758 residue, resulting in its activation. Under hypoxic stress, NAD-dependent protein deacetylase sirtuin-5 promotes the dissociation of PPA2 and NEDD4 by inducing PPA2 desuccinylation at lysine 176, contributing to the improved stability of HIF-1alpha under hypoxic conditions. Our findings reveal a tumor-suppressive role of PPA2 in HIF-1alpha-dependent colorectal cancer, providing a potential therapeutic target and prognostic strategy.

**Keywords** HIF-1α; Hypoxia; Metastatic Colorectal Cancer; NEDD4; PPA2 Succinylation
**Subject Categories** Cancer; Metabolism

## Introduction

Colorectal cancer (CRC) is the third most commonly diagnosed malignancy and the second leading cause of cancer-related death worldwide (Sung et al, 2021). Metastatic colorectal cancer (mCRC) has a high incidence rate and accounts for most CRC-related deaths, with a 5-year survival rate of 91% for localized disease, and 14% for distant disease (Siegel et al, 2023). Approximately 15–30% of CRC patients are diagnosed with mCRC at initial diagnosis, and up to 50% of CRC patients with localized disease eventually develop metastases (Fernández Montes et al, 2023). The liver is the most common metastatic site of CRC, with the lung and peritoneum closely behind it. Despite improvements in surgical methods and comprehensive multidisciplinary care, the efficacy of cytotoxic chemotherapy and targeted therapy remains unsatisfactory in patients with mCRC (Douillard et al, 2010; Heinemann et al, 2014; Mayer et al, 2015; Venook et al, 2017), and the relapse rate is as high as 68% within three years after curative resection of metastatic tumors (Galjart et al, 2016). Therefore, exploring the mechanism underlying CRC metastasis and identifying potential molecular targets for risk evaluation and therapeutic intervention are crucial.

Metastasis is a complex and dynamic process that depends on interactions between tumor cells and microenvironmental factors (Chen et al, 2011; Gao et al, 2012; Sceneay et al, 2012). Hypoxia is a prominent feature of the tumor microenvironment (TME) of 50% to 60% solid tumors and governs the metastatic potential of tumor cells at multiple steps within the metastatic cascade (Barsoum et al, 2014; Eisinger-Mathason et al, 2013; Loo et al, 2015). Mounting studies have provided evidence for the existence of hypoxia in CRC (Goethals et al, 2006; Havelund et al, 2013; Wendling et al, 1984). Under hypoxic conditions, cellular physiological activities are reduced to avoid further increasing oxygen consumption and maintain basic functions of cells and tissues. Cancer cells adapt to hypoxia by utilizing metabolic reprogramming to refuel various malignant behaviors, including migration and invasion, which is an innate advantage of tumors (Park et al, 2023; Shigeta et al, 2023; Wise et al, 2011; Wu et al, 2021). Colon cells are exposed to acute hypoxia and undergo a metabolic shift towards increasing glycolysis during initial dissemination to the liver (Loo et al, 2015), which implies that the ability to survive the metabolic stress associated with hypoxia and avoid abnormal apoptosis upon

[1]Department of Gastrointestinal Surgery, The First Affiliated Hospital, Sun Yat-sen University, Guangzhou, Guangdong 510080, China. [2]Molecular Diagnosis and Gene Testing Center, The First Affiliated Hospital, Sun Yat-sen University, Guangzhou, Guangdong 510080, China. [3]Center of Hepato-Pancreato-Biliary Surgery, The First Affiliated Hospital of Sun Yat-Sen University, Guangzhou, Guangdong 510080, China. [4]Organ Transplant Center, The First Affiliated Hospital, Sun Yat-sen University, Guangzhou 510080, China. [5]Department of General Surgery, The First Affiliated Hospital of Nanchang University, Medical College of Nanchang University, Nanchang, Jiangxi 330006, China. [6]Department of Pathology, The First Affiliated Hospital of Sun Yat-sen University, Guangzhou, Guangdong 510080, China. [7]Center for Translational Medicine, The First Affiliated Hospital, Sun Yat-sen University, Guangzhou, Guangdong 510080, China. [8]Department of Gastrointestinal Surgery, Xiang'an Hospital of Xiamen University, School of Medicine, Xiamen University, Xiamen, Fujian 361000, China. [9]These authors contributed equally: Xiang Zhang, Yuqin Di, Youpeng Wang. ✉E-mail: kezunfu@mail.sysu.edu.cn; wangzy256@mail.sysu.edu.cn; wlhe@xah.xmu.edu.cn

entering the circulation is crucial for the successful colonization of disseminated tumor cells. Despite the increased reports on the interplay between hypoxia adaptation, including metabolic reprogramming, and tumor metastasis, there is still a lack of practical clinical applications. Thus, a better understanding of the role of hypoxia in mCRC may reveal promising therapeutic approaches and prognostic markers.

Inorganic pyrophosphatases (PPAs) are pivotal enzymes that catalyze the intracellular hydrolysis of pyrophosphate (PPi). Humans have two variants of PPA that share sequence homology, PPA1 and PPA2. PPA1 is highly expressed in various tumors, facilitates malignant behaviors, and correlates with unfavorable outcomes (Guo et al, 2021; Li et al, 2017; Niu et al, 2017; Yang et al, 2015). Low PPA2 expression correlates with poor prognosis in kidney renal clear cell carcinoma (Zhu et al, 2021). However, the specific role of PPA2 in other types of cancer, including CRC, remains unknown. Here, we identified PPA2 as a favorable prognostic factor in CRC, and revealed that the deficiency of PPA2 in CRC contributes to metastasis in a hypoxia-inducible factor 1 alpha (HIF-1α)-dependent manner. PPA2 facilitates ubiquitin-mediated proteasomal degradation of HIF-1α by recruiting E3 ligase neural precursor cell-expressed developmentally downregulated gene 4 (NEDD4) and promoting NEDD4 dephosphorylation at threonine 758 (T758) residue. Moreover, we demonstrated that PPA2 is desuccinylated at lysine 176 (K176) residue by NAD-dependent protein deacetylase sirtuin-5 (SIRT5) under hypoxic stress, which weakens its ability to recruit and dephosphorylate NEDD4, thereby increasing the stability of HIF-1α under hypoxic conditions.

## Results

### PPA2 downregulation correlates with increased CRC metastasis and poor prognosis

To identify the critical metabolic enzymes involved in suppressing CRC progression, we used a systematic screening strategy. Two Gene Expression Omnibus (GEO) datasets (Data ref: Sheffer et al, 2012; Data ref: Solé et al, 2014) revealed that 33 genes encoding metabolic enzymes were differentially expressed in CRC between primary tumor tissues (T) and normal tissues (N) and between liver metastasis tissues (M) and primary tumor tissues ($\log_2 FC < -0.40$, adj. $P < 0.05$). Interestingly, only five genes were identified as suppressive candidate genes in mCRC via univariate cox proportional hazards analysis in the Cancer Genome Atlas (TCGA) dataset (HR < 1.0, $P < 0.05$) (Fig. 1A). The details of the screening strategy are provided in Dataset EV1. Among the five genes, deficiencies in SDHB and GSR have been reported to contribute to CRC metastasis (Gu et al, 2020; Wang et al, 2016). Therefore, we individually silenced the expression of endogenous PPA2, ELOVL6, or DLAT in DLD1 cells using specific small interfering RNAs (siRNAs) (Fig. EV1A). Transwell assays revealed more pronounced promotion of migration and invasion after PPA2 knockdown compared to ELOVL6 or DLAT knockdown in DLD1 cells (Figs. 1B and EV1B). Thus, we focused on studying the function of PPA2 in tumor metastasis. Moreover, we observed consistently low PPA2 expression in advanced-stage patients, particularly those with lymphatic or distant metastasis, within a GEO dataset of CRC (Data ref: Sheffer et al, 2012) (Fig. 1C). RNA-seq data for rectum adenocarcinoma (READ) and colon adenocarcinoma (COAD)

from the TCGA database also confirmed that reduced PPA2 expression was correlated with advanced tumor stages (Fig. EV1C,D). Furthermore, analysis of the TCGA pan-cancer dataset obtained from the UCSC Xena project (Goldman et al, 2020) revealed that PPA2 expression was downregulated exclusively in CRC (including COAD and READ) compared to corresponding normal tissues, but not in other cancer types (Fig. 1D). These results imply a potential role for PPA2 in inhibiting CRC metastasis.

To validate our bioinformatics findings, we performed qRT-PCR, immunoblotting (IB), and immunohistochemistry (IHC) analyses to examine PPA2 expression in paired CRC tumor and adjacent peritumoral tissues, and showed lower PPA2 expression in CRC tissues (Fig. 1E–G). In addition, we also compared PPA2 expression in the primary CRC tumors of different stages. The results showed that CRC patients of advanced stages had much lower protein levels of PPA2 than those of early stages (Fig. 1H). Similarly, reduced PPA2 level was also observed in primary tumor tissues from CRC patients with metastasis compared to those without (Fig. 1I). These results strongly support that the reduced expression of PPA2 is correlated with disease progression of CRC.

In addition, Kaplan-Meier survival analysis revealed that patients with low PPA2 expression exhibited shorter survival time compared to those with high PPA2 expression in all stages CRC from the Sun Yat-sen University First Affiliated Hospital (SYSU-FAH) CRC dataset, TCGA COAD and READ datasets (Fig. 1J–L). Interestingly, low PPA2 was correlated with a poor prognosis for tumors in advanced stages (III + IV) (Fig. EV1E,F), but not those in early stages (I + II) (Fig. EV1G,H). Collectively, our findings suggest that PPA2 deficiency facilitates tumor cell migration and invasion in vitro, and decreased PPA2 is associated with mCRC, leading to a poor prognosis at advanced stages of CRC.

### PPA2 inhibits tumor cell migration and hypoxic resistance in vitro and suppresses CRC metastasis in vivo

To explore and validate the function of PPA2 in cell phenotype, we designed two independent short hairpin RNAs (shRNAs) that target PPA2 (shPPA2) to stably knockdown its expression in DLD1 and SW1116 cells (Fig. 2A). RNA-seq analysis of PPA2-depleted DLD1 cells revealed significant enrichment of the HIF-1 signaling pathway (Fig. EV2A), suggesting a potential link between PPA2 and cellular response to hypoxic stress, a critical factor in tumor metastasis. Hence, we sought to determine the relationship between the function of PPA2 and hypoxic stress in tumor metastasis. Transwell migration and invasion assays under normoxia and hypoxic conditions demonstrated that PPA2 depletion significantly enhanced the migration and invasion of tumor cells under hypoxic stress (Fig. 2B,C). These findings were further corroborated by wound-healing assays in DLD1 and SW1116 cells (Fig. EV2B). Given that CRC cells often encounter hypoxic conditions during the metastatic process, such as acute hypoxia that may occur when passing through the hypoxemic portal vein and during early colonization of the hypoxic liver microenvironment, which results in abnormal death in tumor cells in this process (Loo et al, 2015), we investigated the impact of PPA2 depletion on cell viability and apoptosis under hypoxia. The results revealed that PPA2-depleted cells exhibited increased cell viability and decreased apoptosis rates under continuous hypoxia compared to control cells (Figs. 2D,E and EV2C). In contrast, no difference in baseline cell viability and apoptosis was observed under normoxia.

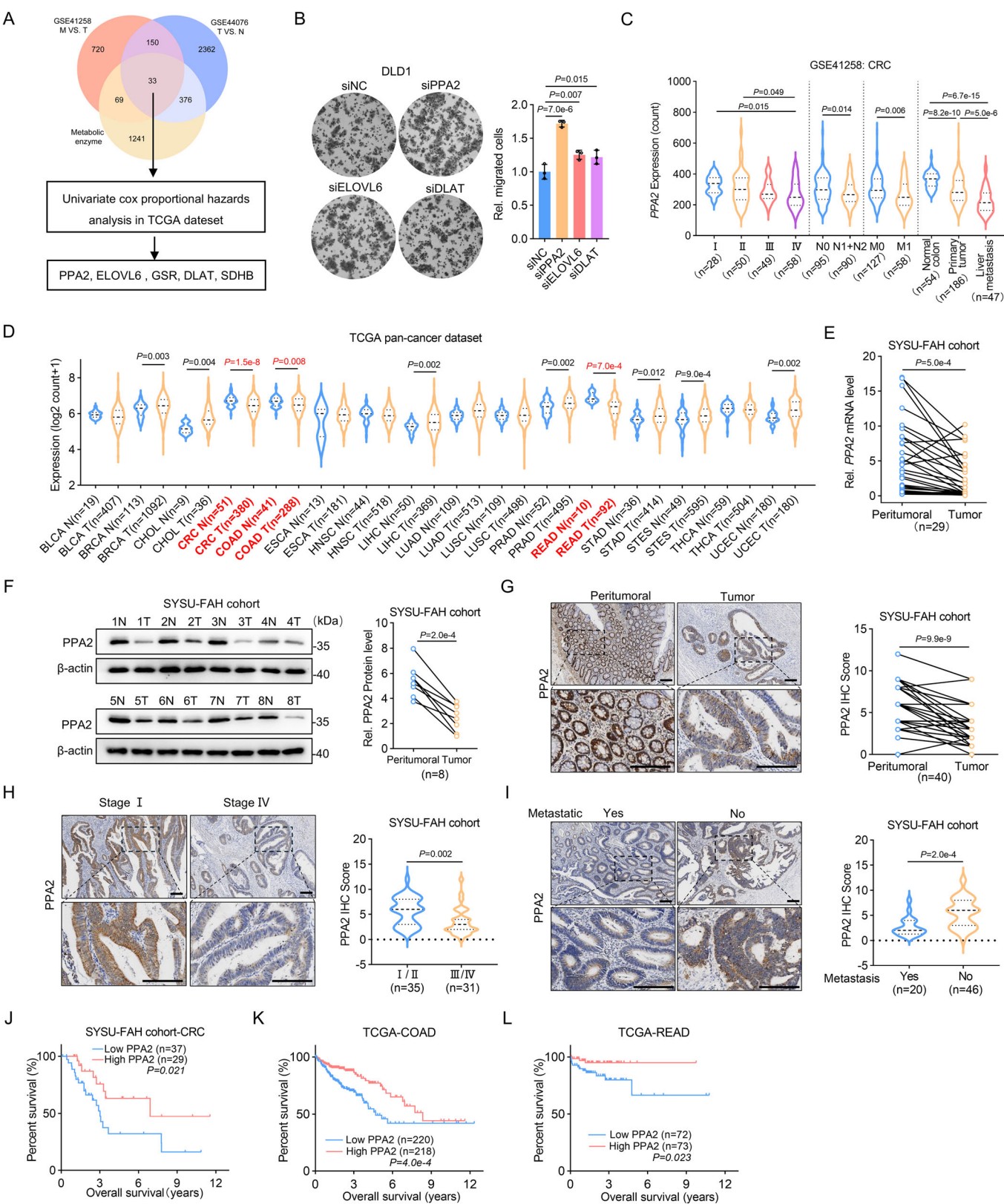

**Figure 1.  PPA2 downregulation correlates with increased CRC metastasis and poor prognosis.**

(**A**) Venn diagrams showed overlapped potential suppressive targets of mCRC. (**B**) PPA2, ELOVL6, or DLAT was individually knocked down using siRNA in DLD1 cells and subsequently used for transwell migration assay. (**C**) GSE41258 RNA-seq data of CRC patients were analyzed. *PPA2* mRNA levels were compared between indicated groups. (**D**) TCGA RNA-seq data of multiple cancer types were analyzed. *PPA2* mRNA levels were compared between tumors and normal tissues. (**E, F**) *PPA2* mRNA (**E**) and protein (**F**) levels in tumor and adjacent peritumoral tissues. (**G–I**) PPA2 protein expression on the tumors and peritumoral tissues (**G**), early stages and advanced stages tissue (**H**), and on the tumors with or without metastatic recurrence (**I**). Of note is that some of the dots on the graphs represent more than one sample (i.e., some scores overlapped). Scale bar, 200 μm. (**J–L**) Kaplan–Meier analysis of survival durations based on the PPA2 expression status in CRC patients from SYSU-FAH (**J**), TCGA-COAD (**K**) and TCGA-READ (**L**) cohorts. Data information: Data are mean ± SD from three biological replicates (**B**). Statistics: one-way ANOVA (**B, C**); unpaired two-tailed student's t-test (**C, D, H, I**); paired two-tailed student's t-test (**E–G**); log-rank test (**J–L**). Source data are available online for this figure.

To further explore the functional role of PPA2 in CRC metastasis, we conducted gain-of-function assays in which PPA2 was overexpressed in DLD1 and SW1116 cells (Fig. 2F). Overexpression of PPA2 in DLD1 and SW1116 cells significantly suppressed their migration and invasion abilities, especially under hypoxic conditions (Fig. 2G,H). This overexpression also reduced cell viability and increased cell apoptosis after continuous hypoxic stress (Fig. 2I–L). To determine whether the function of PPA2 relies on its enzymatic activity, we restored PPA2 expression in PPA2-depleted cells using either shPPA2-resistant wild-type (rPPA2 WT) or enzymatically dead PPA2 mutant (ED: E172K) (Kennedy et al, 2016) (Fig. EV2D). While restoration with rPPA2 WT attenuated the migration and invasion induced by PPA2 depletion, restoration with rPPA2 ED did not have the same effect (Fig. EV2E–G). Similarly, rPPA2 WT, not rPPA2 ED, abrogated the increased hypoxic resistance observed in PPA2-depleted cells (Fig. EV2H). These findings demonstrate that the enzymatic activity of PPA2 is critical for its anti-metastatic effects.

To validate the impact of PPA2 on tumor metastasis in vivo, we established a mouse xenograft model of liver metastasis by splenic injection of PPA2-depleted or PPA2-overexpressed DLD1 cells into BALB/c-nude mice. Mice were sacrificed after 6 weeks, and the CRC liver metastasis (CRLM) burden was evaluated according to the number and maximum diameter of metastatic tumors. Our findings demonstrated that the CRLM burden was significantly increased by PPA2 knockdown (Fig. 2M). Notably, while the overexpression of PPA2 WT significantly inhibited the CRLM burden, the overexpression of PPA2 ED did not have this effect (Fig. 2N), suggesting that the in vivo anti-metastatic effects of PPA2 also depend on its enzymatic activity. These results suggest that PPA2 knockdown promotes tumor cell migration, invasion, and hypoxic resistance in vitro and facilitates tumor metastasis in vivo, possibly through the activation of the HIF-1 signaling pathway. Conversely, forced expression of PPA2, in an enzymatic activity-dependent manner, inhibits these processes, highlighting the crucial role of PPA2 in regulating CRC metastasis under hypoxic conditions.

## PPA2 inhibits glycolysis-mediated CRC metastasis by facilitating the degradation of HIF-1α

To investigate how PPA2 responds to hypoxic stress, we performed RNA-seq on control and PPA2-depleted DLD1 cells exposed to hypoxia. As expected, gene set enrichment analysis (GSEA) and heatmap analysis demonstrated that significant upregulation of HIF-1α downstream target genes in hypoxic PPA2-depleted cells, including known metastasis modulators like *LOX* and *SNAI1*, as well as

metabolic enzymes involved in glycolysis (Figs. 3A and EV3A). The differentially expressed HIF-1α target genes were confirmed using qRT-PCR in CRC cells and liver metastatic tissues of previously established mouse models (Figs. 3B,C and EV3B,C). Furthermore, we examined whether PPA2 affected glycolysis in CRC cells. Systematic analysis of energy metabolomics revealed that PPA2 deficiency prominently promoted glycolysis of DLD1 cells under hypoxic stress (Fig. 3D), and lactate was identified as a significantly increased metabolite in the PPA2 depletion group (Fig. 3E). Metabolite detection in the media confirmed that PPA2 knockdown facilitated glucose consumption and lactate production in DLD1 cells upon stimulation with hypoxic stress (Fig. 3F,G). IOX2, a chemical inhibitors of prolyl hydroxylase domain 2 (PHD2)/ Von Hippel–Lindau (VHL) system, was applied to mimic hypoxic conditions in the extracellular acidification rate (ECAR) analysis, and the results further confirmed that PPA2 depletion in combination with IOX2 treatment significantly increased glycolysis and glycolytic capacity of DLD1 cells (Fig. 3H–J). Conversely, overexpression of PPA2 exhibited an opposite effect on glycolysis of DLD1 cells (Fig. EV3D–H). These results reveal that PPA2 inhibits the metabolic rewiring of tumor cells toward a glycolytic phenotype in response to hypoxic stress, thereby creating a barrier for tumor cells in their adaptation to hypoxia and successful metastasis.

Given that PPA2 responds to hypoxic stress and is involved in modulating the HIF-1 signaling pathway, we examined the effect of PPA2 on HIF-1α level. IB analysis revealed that, compared with control cells, PPA2 depletion resulted in a dramatic increase in HIF-1α after hypoxic stress stimulation and a slight increase under normoxia (Fig. 3K). In contrast, the overexpression of PPA2 reduced the HIF-1α protein levels after hypoxia treatment (Fig. EV3I). Hypoxia-induced upregulation of HIF-1α is primarily attributed to posttranslational mechanisms. PPA2 induces rapid changes of HIF-1α in response to short-term hypoxic stress, indicating that PPA2 may interfere with the stability of HIF-1α. As expected, PPA2 knockdown or overexpression did not affect HIF-1α transcription (Fig. EV3J,K). Meanwhile, PPA2 knockdown prolonged the half-life of the HIF-1α protein under cycloheximide (CHX) treatment conditions (Fig. 3L). Moreover, treatment with the proteasome inhibitor MG132 abrogated the increase in HIF-1α levels induced by PPA2 depletion under both normoxia and hypoxia (Fig. 3M), indicating that PPA2 deficiency-induced increase in HIF-1α protein level was due to impaired proteasomal degradation. To further clarify the impact of PPA2 on the HIF-1 signaling pathway, we employed a luciferase reporter plasmid containing a conventional hypoxia-response element (HRE). PPA2 knockdown significantly increased the luciferase activity of the HRE reporter, especially under hypoxic conditions (Fig. 3N). In addition, chromatin immunoprecipitation (ChIP)-qPCR analysis

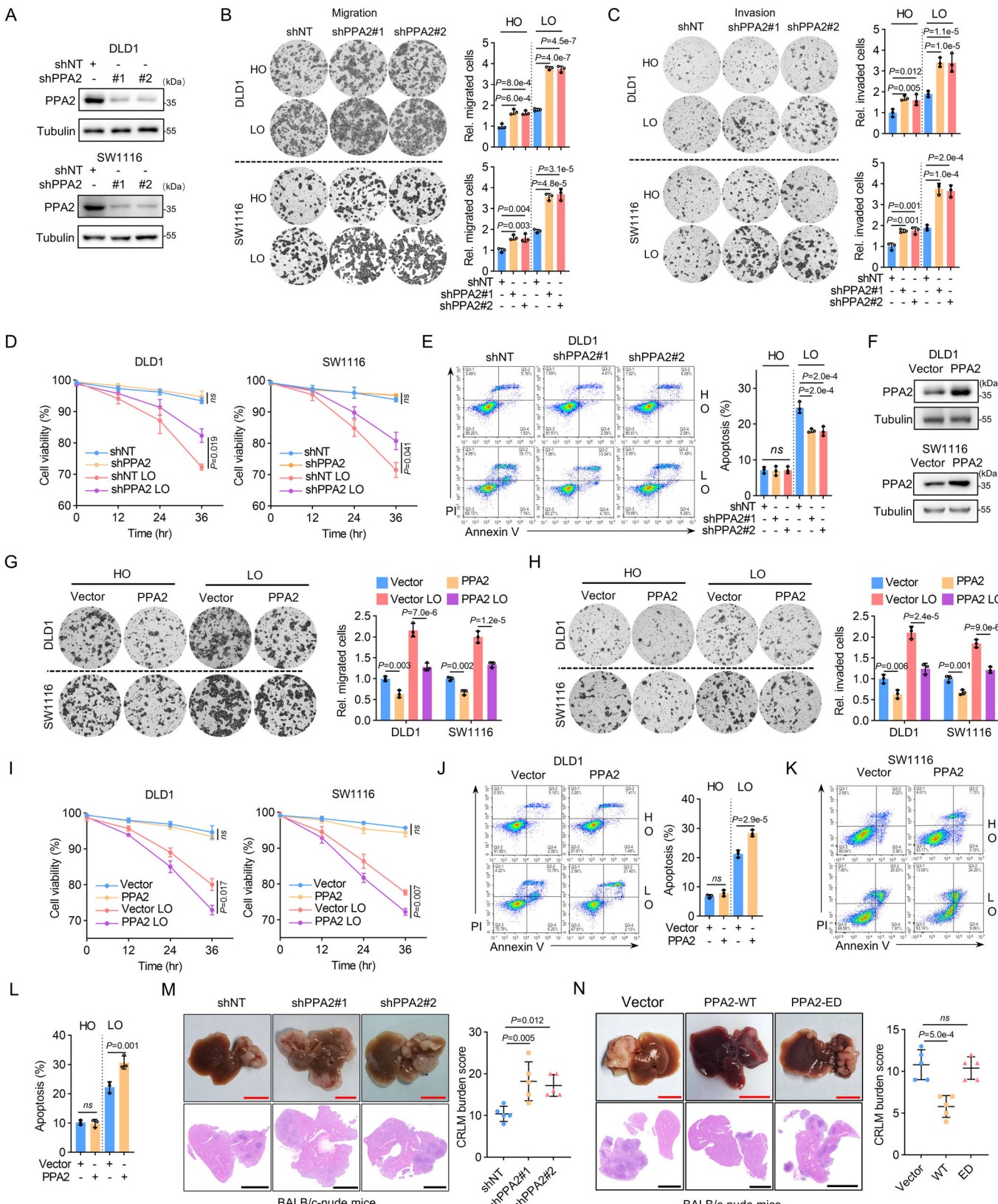

**Figure 2. PPA2 inhibits tumor cell migration and hypoxic resistance in vitro and suppresses CRC metastasis in vivo.**

Normoxia (20% oxygen) is also termed HO. Hypoxia (1% oxygen) is also termed LO. (A) IB analysis in DLD1 and SW1116 cells expressing shNT or shPPA2. (B, C) Control or PPA2-depleted DLD1 and SW1116 cells were used for transwell migration (B) and invasion (C) assays under normoxia or hypoxia, respectively. (D, E) Control or PPA2-depleted cells were collected after incubation under normoxia or hypoxia. Cell viability was assessed by trypan blue staining (D), and PI/Annexin V double staining was then used to measure the percentage of apoptotic cells via flow cytometry (E). (F) IB analysis in DLD1 and SW1116 cells overexpressing vector or PPA2. (G, H) DLD1 and SW1116 cells overexpressing vector or PPA2 were used for transwell migration (G) and invasion (H) assay under normoxia or hypoxia, respectively. (I–L) DLD1 and SW1116 cells overexpressing vector or PPA2 were collected after incubation under normoxia or hypoxia, respectively. Cell viability was assessed by trypan blue staining (I), and PI/Annexin V double staining was used to measure the percentage of apoptotic cells via flow cytometry (J–L). (M, N) Liver metastasis mouse model from splenic injection using the indicated DLD1 cells. Representative images of the liver and HE-stained liver section were shown, and the CRLM burden was calculated in each group. Red scale bar, 1 cm; black scale bar, 5 mm. Data information: Data are mean ± SD from three biological replicates (B–E, G–L) and five biological replicates (M, N). Statistics: one-way ANOVA (B, C, E, G, H, J, L–N); two-way ANOVA (D, I). Source data are available online for this figure.

further validated that the binding of HIF-1α to the promoter region of its downstream target genes was significantly increased by PPA2 depletion under hypoxic stress (Fig. EV3L,M). These results suggest that PPA2 negatively regulates the stability of HIF-1α and the expression of its downstream target genes.

Next, we knocked out HIF-1α to investigate whether the function of PPA2 depends on HIF-1α. HIF-1α knockout effectively diminished the levels of hypoxic HIF-1α and its downstream metastasis and glycolysis target genes in both PPA2-depleted and control cells (Figs. 3O,P and EV3N). Moreover, HIF-1α knockout counteracted PPA2 knockdown-induced cell migration and invasion under hypoxia (Figs. 3Q,R and EV3O), and abolished hypoxic resistance induced by PPA2 deficiency, as evidenced by decreased cell viability in HIF-1α knockout groups (Fig. EV3P). Consistently, HIF-1α knockout not only suppressed glucose consumption and lactate production as expected, but also abolished the effect of PPA2 depletion or overexpression on metabolites level (Figs. 3S,T and EV3Q,R). These findings were corroborated in an in vivo study, where the increased CRLM burden observed in PPA2-depleted cells was significantly attenuated by HIF-1α knockout (Fig. 3U). Together, these results indicate that the function of PPA2 in CRC depends on HIF-1α. Specifically, PPA2 negatively regulates HIF-1α stability through proteasomal degradation, ultimately influencing HIF-1α-mediated cellular processes, including metastasis, glycolysis, and hypoxic survival.

## PPA2 recruits and activates NEDD4 for ubiquitin-mediated proteasomal degradation of HIF-1α

As PPA2 regulates HIF-1α by modulating proteasomal degradation, we examined whether PPA2 affects HIF-1α ubiquitination. Immunoprecipitation (IP) analysis revealed that PPA2 significantly promoted the ubiquitination of HIF-1α under normoxic and hypoxic conditions (Fig. 4A). While PHD/VHL system-mediated ubiquitination is crucial for HIF-1α degradation under normoxic conditions (Cockman et al, 2000), we observed that PPA2 did not affect the interaction between VHL and HIF-1α (Fig. EV4A). In addition, PHD2 inhibitor IOX2-induced HIF-1α was also significantly modulated by PPA2 knockdown or overexpression (Fig. EV4B). Consistently, the genetic inhibition of VHL stabilized HIF-1α as expected, but did not abolish the effect of PPA2 on the protein and ubiquitination level of HIF-1α (Fig. EV4C,D). Moreover, the interaction between PPA2 and HIF-1α was confirmed by co-IP analysis; however, hypoxia treatment partially weakened this interaction (Fig. EV4E). These results suggest that PPA2 regulates HIF-1α stability through E3 ubiquitin ligases other than the PHD/VHL

system. To identify potential non-canonical E3 ligases, we analyzed the E3 ligases of HIF-1α predicted by the UbiBrowser database (http://ubibrowser.bio-it.cn/ubibrowser/) and performed molecular docking analysis with PPA2 via ZDOCK and HDOCK. This analysis identified five candidate E3 ligases with a high likelihood of interacting with PPA2 (Fig. EV4F, Dataset EV2). Next, siRNAs targeting the five candidate E3 ligases were individually used to knock down these genes in PPA2-overexpressing DLD1 cells (Fig. EV4G). The results revealed that only the knockdown of NEDD4 fully rescued PPA2-induced decrease in hypoxic HIF-1α protein level (Fig. EV4H). NEDD4 is one member of the NEDD4 subfamily, which consists of nine members in humans, each having overlapped yet distinct repertoires of binding partners/substrates. To determine whether multiple NEDD4 family members can bind to HIF-1α, we performed exogenous co-IP in HEK293T cells. The results revealed that only WWP2, NEDD4, and NEDD4L were notably associated with HIF-1α, while the other six members did not show this interaction (Fig. EV4I). Additionally, knockdowns of WWP2 or NEDD4L did not fully rescue the PPA2-induced decrease in hypoxic HIF-1α protein levels as effectively as NEDD4 did; instead, they resulted in only a moderate increase in HIF-1α (Fig. EV4G,H), corroborating previous findings by Liu et al (2023). Collectively, these results suggest that PPA2 relies on NEDD4 for the degradation of HIF-1α. Further analysis showed that NEDD4 knockdown significantly decreased the level of ubiquitinated HIF-1α (Fig. 4B). More importantly, NEDD4 knockdown abrogated the PPA2-induced increase in HIF-1α ubiquitination under both normoxic and hypoxic conditions (Figs. 4C and EV4J). Co-IP results revealed that the interaction between NEDD4 and HIF-1α was robust under normoxia but considerably weakened under hypoxic conditions (Fig. 4D). Notably, PPA2 strengthened this interaction under both normoxic and hypoxic conditions (Fig. 4D); however, it did not influence the association between HIF-1α and WWP2 or NEDD4L (Fig. EV4K), further excluding the possibility that WWP2 or NEDD4L is involved in the regulation of HIF-1α by PPA2. Furthermore, the interaction between PPA2 and HIF-1α was impaired by both NEDD4 knockdown and hypoxic stress, either individually or in combination (Fig. 4E). Moreover, both endogenous and exogenous co-IP confirmed the association between PPA2 and NEDD4, and this interaction was attenuated by hypoxic stress (Figs. 4F and EV4L). Similar findings were observed via immunofluorescence (IF) colocalization analysis (Fig. 4G). Next, we investigated the possible mechanism by which PPA2 affected NEDD4 function. Phosphorylation-induced inactivation of NEDD4L, another member of the NEDD4 family, has been reported (Arroyo et al, 2011). Given that PPA1, another member of the PPA family, has been shown to directly bind and dephosphorylate its substrate (Wang et al, 2017), we hypothesized that PPA2 might dephosphorylate NEDD4. IP analysis revealed that the threonine

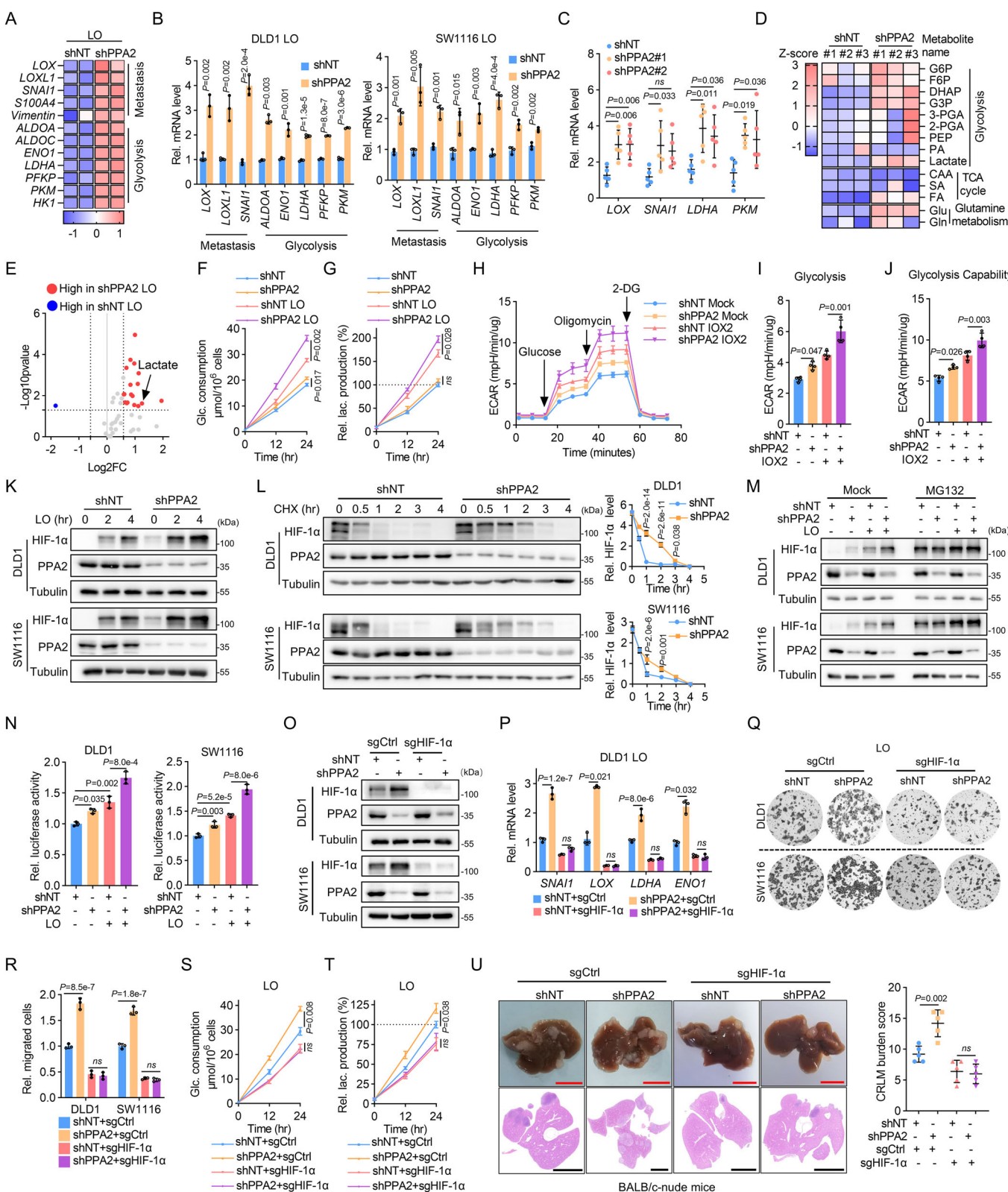

phosphorylation level of NEDD4 gradually decreased with increasing amounts of PPA2 transfection (Fig. 4H). Based on these results, we speculated that PPA2 recruits and activates NEDD4 via its phosphatase activity, and then the PPA2-NEDD4 E3 ligase complex subsequently

recognizes and binds to HIF-1α, thereby facilitating its ubiquitin-mediated proteasomal degradation. To validate this hypothesis, we constructed NEDD4 truncations and found that PPA2 interacted with amino acids (AAs) 600-930 of NEDD4, which contains four WW

**Figure 3. PPA2 inhibits glycolysis-mediated CRC metastasis by facilitating the degradation of HIF-1α.**

(A) Control or PPA2-depleted DLD1 cells were incubated under hypoxia for 12 h and were collected for RNA-seq analysis. The upregulated HIF-1α target genes in PPA2-depleted DLD1 cells are shown in a heatmap. (B) Control or PPA2-depleted DLD1 and SW1116 cells were collected after incubation under hypoxic conditions for 12 h, and the mRNA levels of HIF-1α target genes were analyzed by qRT-PCR. (C) The mRNA level of HIF-1α target genes was analyzed via qRT-PCR in liver metastasis tissues derived from splenic injection using PPA2-depleted DLD1 cells. (D, E) Control or PPA2-depleted DLD1 cells were incubated under hypoxia for 12 h and then collected for metabolomics analysis. The relative abundance of metabolites involved in glycolysis, TCA cycle, and glutamine pathway is displayed by the heatmap (D) and volcano plot (E). D-Glucose-6-phosphate, G6P; D-Fructose-6-phosphate, F6P; Dihydroxyacetone-phosphate, DHAP; Glyceraldehyde-3-phosphate, G3P; 3-phosphoglycerate, 3-PGA; 2-Phospho-D-glycerate, 2-PGA; Phosphoenolpyruvic-acid, PEP; Pyruvic-acid, PA; Cis-Aconitic-acid, CAA; Succinic-Acid, SA; Fumaric-acid, FA; L-Glutamic-acid, Glu; Glutamine, Gln. (F, G) The media of control or PPA2-depleted DLD1 cells were collected after incubation under normoxia or hypoxia for the indicated time to detect the glucose (F) and lactate levels (G). The lactate level of shNT group at 24 h was used as the standard for data normalization in (G). (H–J) Control or PPA2-depleted DLD1 cells were used for ECAR experiments with or without IOX2. (K) Control or PPA2-depleted DLD1 and SW1116 cells were collected after incubation under normoxia or hypoxia induction for the indicated time. (L) Control or PPA2-depleted DLD1 and SW1116 cells were cultured under hypoxia with MG132 for 4 h, followed by treatment with CHX for the indicated time. The intensity of HIF-1α expression was quantified by Image J software and normalized to Tubulin. (M) Control or PPA2-depleted DLD1 and SW1116 cells were collected after incubation under normoxia or hypoxia for 4 h with or without MG132. (N) Luciferase assay of HRE-luciferase reporter plasmid in control or PPA2-depleted DLD1 and SW1116 cells cultured under normoxia or hypoxia for 12 h. (O) Control or PPA2-depleted DLD1 and SW1116 cells treated with or without sgRNA targeting HIF-1α were collected after incubation under hypoxia for 4 h. (P) DLD1 cells in (O) were collected after incubation under hypoxia for 12 h to detect the mRNA level of HIF-1α target genes via qRT-PCR. (Q, R) Cells in (O) were used for transwell migration assay under hypoxia. (S, T) The media of DLD1 cells in (O) were collected after incubation under hypoxia for the indicated time to detect the glucose (S) and lactate levels (T). The lactate level of shNT+sgCtrl group at 24 h was used as the standard for data normalization in (T). (U) Liver metastasis mouse model from splenic injection using DLD1 cells in (O). Representative images of the liver and HE-stained liver section were shown, and the CRLM burden was calculated in each group. Red scale bar, 1 cm; black scale bar, 5 mm. (K–M, O) IB analyses were performed using the indicated antibodies. Data information: Data are mean ± SD from three biological replicates (B, D–G, N, P–T), four biological replicates (H–J), three technical replicates (L) and five biological replicates (C, U). Statistics: unpaired two-tailed student's t-test (B, E), one-way ANOVA (C, I, J, N, P, R, U); two-way ANOVA (F, G, L, S, T). Source data are available online for this figure.

domains responsible for association with its target protein (Fig. 4I,J). We identified four potential threonine phosphorylation residues within AAs 600-930 of NEDD4 using PhosphoSitePlus (https://www.phosphosite.org/) and mutated to alanine (A) to generate a phosphorylation-deficient mutant. Only the T758A significantly abrogated PPA2-knockdown-induced NEDD4 phosphorylation (Figs. 4K and EV4M). Further corroborating these results, the computational model of the structure predicted by the ZDOCK and PyMOL software revealed that T758 was within the NEDD4/PPA2 interaction region (Fig. 4L). Moreover, NEDD4 T758 was proved to be conserved across multiple species (Fig. EV4N).

To investigate the impact of NEDD4 phosphorylation at T758 on its ubiquitination and protein degradation activities, we performed co-IP assays. These assays revealed that hypoxia failed to induce the dissociation of NEDD4 T758A from HIF-1α, unlike the NEDD4 WT (Fig. 4M). Consistently, the NEDD4 T758A mutant increased the ubiquitinated forms of HIF-1α under hypoxia compared to NEDD4 WT (Fig. 4N). These results collectively suggest that PPA2 dephosphorylates NEDD4 at T758, activating NEDD4 to promote ubiquitin-mediated proteasomal degradation of HIF-1α.

## SIRT5-mediated PPA2 desuccinylation at K176 impairs PPA2 function

Next, we explored the mechanism underlying the dissociation of PPA2 from NEDD4 under hypoxic stress. Posttranslational modifications (PTMs) of lysine are crucial for various cellular signaling events (Khan et al, 2014; Park et al, 2013; Zhang et al, 2019). We found that PPA2 exhibited detectable levels of lysine acetylation and methylation, along with a lower level of lactylation, while malonylation was scarcely detected. No differences in these modifications were observed under hypoxic treatment. In contrast, PPA2 showed notable lysine succinylation modification under normoxic conditions, which gradually decreased with prolonged hypoxic treatment in both DLD1 and SW1116 cells. (Fig. 5A,B). An

in vitro succinylation assay confirmed that succinylation of PPA2 was increased by succinyl-CoA in a dose-dependent manner (Fig. 5C), suggesting that PPA2 can be succinylated. To verify the residues involved in PPA2 succinylation, we analyzed succinylated peptides from DLD1 cells under normoxic or hypoxic conditions using liquid chromatography-mass spectrometry (LC-MS). This analysis identified four lysine residues (K73, K176, K203, and K216) that might undergo hypoxia-induced desuccinylation of PPA2 (Fig. EV5A). To mimic the desuccinylated state, these residues were subsequently mutated to arginine (R) individually as previously described (Tong et al, 2021). Intriguingly, while the succinylation levels of PPA2 mutants K73R, K203R, and K216R were reduced under hypoxia, the succinylation level of K176R mutant remained unchanged (Figs. 5D and EV5B). To further confirm this succinylated site in PPA2, we generated a specific antibody against suK176-PPA2. The specificity of this antibody was validated by its special recognition of suK176 in PPA2 WT but not in PPA2 K176R mutant (Fig. 5E). Importantly, we observed that the level of suK176 decreased in a time-dependent manner under hypoxic stress, indicating that K176 is a critical site that is directly desuccinylated in hypoxic conditions (Fig. 5E). Furthermore, the conservation of PPA2 K176 across various species highlights the potential significance of this residue in regulating PPA2 function (Fig. EV5C).

We next examined the impact of PPA2 succinylation at K176 on its function. Co-IP analysis revealed that the interaction between rPPA2 K176R and NEDD4 was weaker than that of rPPA2 WT under both normoxia and hypoxia, and rPPA2 K176R failed to dephosphorylate NEDD4 (Fig. 5F), suggesting that the succinylation at K176 is critical for the interaction between PPA2 and NEDD4. To further probe the role of K176 succinylation, we mimicked the succinylated state by mutating K176 to glutamate (E) (rPPA2 K176E) as previously described (Zhang et al, 2024), and restored PPA2 expression in PPA2-depleted cells. Under hypoxic stress, decreased HIF-1α protein level and increased ubiquitination of HIF-1α were observed in the rPPA2 K176E group compared to

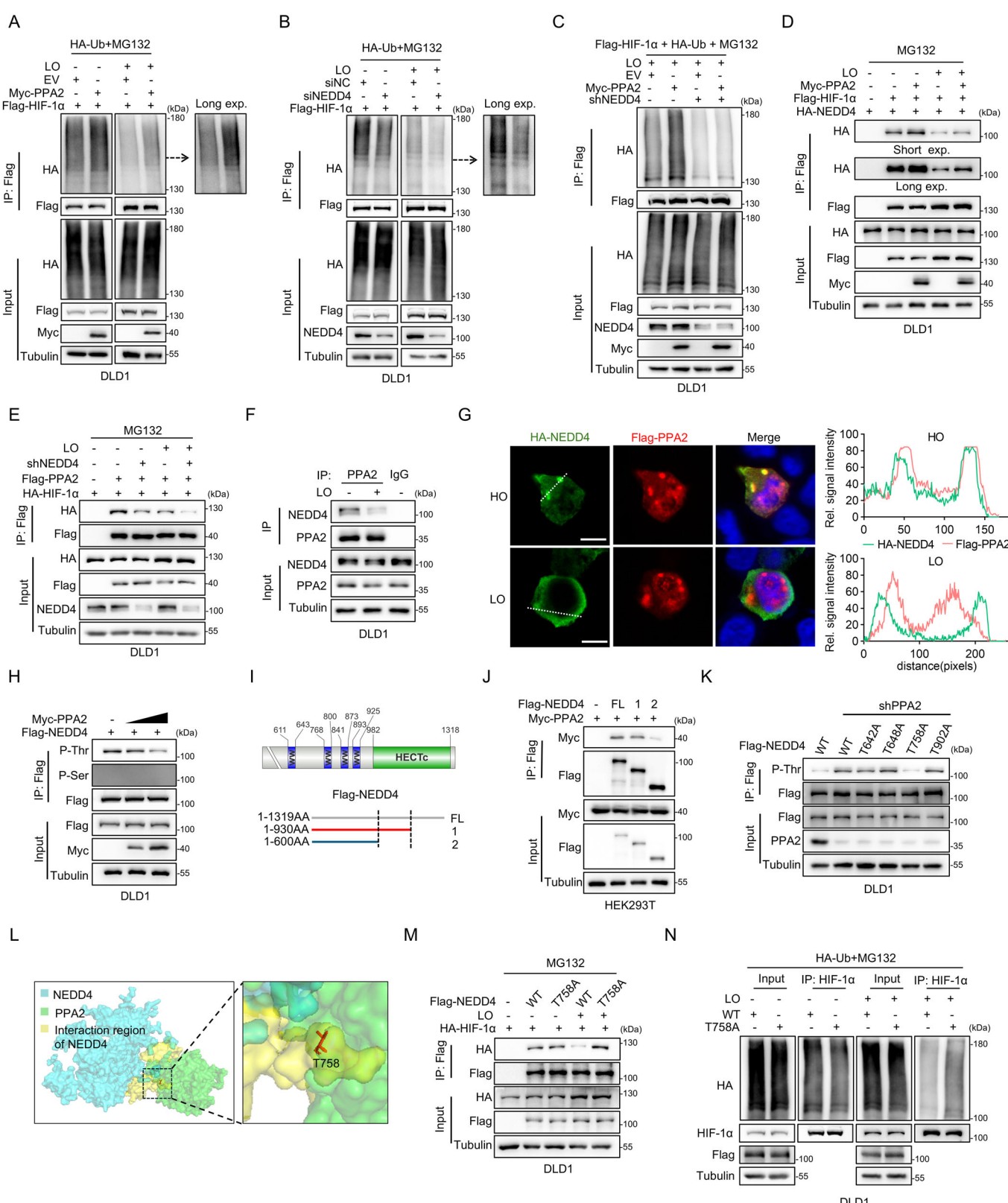

**Figure 4. PPA2 recruits and activates NEDD4 for ubiquitin-mediated proteasomal degradation of HIF-1α.**

(A) DLD1 cells were transfected with Flag-HIF-1α, HA-Ub, and either empty vector (EV) or Myc-PPA2 plasmids. (B) DLD1 cells were transfected with Flag-HIF-1α, HA-Ub, and either siNC or siNEDD4. (C) DLD1 cells stably expressing shNT or shNEDD4 were transfected with Flag-HIF-1α, HA-Ub, and either EV or Myc-PPA2 plasmids. (D) DLD1 cells were transfected with HA-NEDD4 and with or without Flag-HIF-1α and Myc-PPA2 plasmids. (E) DLD1 cells stably expressing shNT or shNEDD4 were transfected with HA-HIF-1α and either EV or Flag-PPA2 plasmids. (F) DLD1 cells were incubated under normoxia or hypoxia and collected for IP analysis. (G) IF showed the colocalization of HA-NEDD4 and Flag-PPA2 under normoxia and the dissociation under hypoxia in DLD1 cells. The fluorescence intensity distribution was quantified using Image J software. (H) DLD1 cells were transfected with Flag-NEDD4 and either EV or increasing amounts of Myc-PPA2 plasmids. (I) Upper panel: structural schematic of NEDD4; lower panel: truncation diagram of NEDD4. (J) DLD1 cells were transfected with Myc-PPA2 and EV, Flag-NEDD4-FL (full length), Flag-NEDD4-1 (AAs: 1–930) or Flag-NEDD4-2 (AAs: 1–600). (K) DLD1 cells stably expressing shNT or shPPA2 were transfected with Flag-NEDD4 WT, T642A, T648A, T758A, or T902A plasmids, respectively. (L) A binding structure model of the PPA2-NEDD4 complex was predicted by ZDOCK and PyMOL software. (M) DLD1 cells were transfected with HA-HIF-1α and EV, Flag-NEDD4 WT, or T758A plasmids. (N) DLD1 cells were transfected with HA-Ub and either Flag-NEDD4 WT or T758A plasmids. (A–F, H, J, K, M, N) IB analyses and IP were performed using the indicated antibodies. Six hours of MG132 (5 μM) treatment was administrated as indicated. Hypoxic stress was administrated in the last 2 h before collection of cells, as indicated. Source data are available online for this figure.

the rPPA2 WT group (Figs. 5G and EV5D,E), indicating that the K176E mutation represents an active form of PPA2. In support of this observation, compared with forced expression of rPPA2 WT, rPPA2 K176E effectively attenuated PPA2-depletion induced HIF-1α downstream target genes (Fig. EV5F), glucose consumption (Fig. EV5G) and lactate production under hypoxic stress (Fig. EV5H). To determine whether K176 succinylation directly affects the PPA2 activity, we performed an in vitro PPA activity assay. Compared with PPA2 WT, the enzymatic dead form of PPA2 (E172K) showed significantly impaired PPA activity, whereas no alteration was detected in the K176R/E mutant (Fig. 5H). However, an in vitro dephosphorylation assay confirmed that neither PPA2 E172K nor K176R could dephosphorylate NEDD4 and produce free phosphate (Fig. 5I). The ability of the PPA family to hydrolyze PPi or dephosphorylate its protein substrate has been reported to be modulated by the same active site of the enzyme (Tezuka et al, 2013), which may explain why the ability of PPA2 in CRC relies on its enzymatic activity, as previously proven. In comparison, we suspected that the K176 succinylation state might affect its role in CRC simply by hindering its recruitment and association with NEDD4 instead of directly affecting its enzymatic activity. Thus, PPA2 E172 and K176 are indispensable for the function of PPA2 in CRC.

To identify the desuccinylase responsible for PPA2 desuccinylation at K176 under hypoxic stress, the pan-sirtuin inhibitor nicotinamide (NAM) was used. The results showed that NAM effectively restored the suK176 level of hypoxic PPA2 in a time-dependent manner (Fig. 5J), suggesting the involvement of a sirtuin. SIRT5 and SIRT7 are $NAD^+$-dependent protein lysine desuccinylases and have been reported to play critical roles in cellular biology (Li et al, 2016; Teng et al, 2024). Using specific siRNAs targeting SIRT5 and SIRT7 (Fig. EV5I,J), we found that SIRT5 knockdown successfully reversed the hypoxia-induced decrease of suK176 level (Fig. 5K), similar to that of NAM. Furthermore, exogenous and endogenous co-IP analyses confirmed the interaction between PPA2 and SIRT5, which were strengthened after hypoxia treatment (Fig. 5L,M). To investigate the impact of SIRT5 on PPA2, we rescued SIRT5 expression with rSIRT5 WT or H158Y mutant, an enzymatically defective SIRT5 (Teng et al, 2024), in endogenous SIRT5-depleted DLD1 cells. Co-IP analysis revealed that, unlike rSIRT5 WT, H158Y failed to desuccinylate PPA2 at K176 (Fig. 5N). Moreover, the phenomenon of hypoxia-induced dissociation of PPA2 from NEDD4 was abolished in endogenous SIRT5-depleted cells, whereas it was recovered by the

forced expression of rSIRT5 WT but not by the H158Y mutant (Fig. 5O). Similar results were observed for the association between PPA2 and HIF-1α (Fig. 5P). In addition, in endogenous SIRT5-depleted DLD1 cells, PPA2-facilitated ubiquitination of hypoxic HIF-1α was counteracted by rSIRT5 WT, but not by the H158Y mutant (Fig. 5Q). Consistently, rSIRT5 WT attenuated the PPA2-induced decrease in hypoxic HIF-1α protein level, but H158Y did not (Fig. EV5K). These results suggest that succinylation of K176 is crucial for PPA2 to associate with and dephosphorylate NEDD4. Under hypoxic conditions, SIRT5 is recruited to desuccinylate PPA2 at K176, leading to the decreased NEDD4-mediated degradation of HIF-1α.

## PPA2 K176 succinylation is critical for mCRC suppression both in vitro and in vivo

To further determine the role of PPA2 K176 succinylation, we rescued PPA2 expression with rPPA2 WT or K176E in endogenous PPA2-depleted cells. Compared to rPPA2 WT, rPPA2 K176E significantly abrogated PPA2-deficiency-induced tumor cell migration and invasion under hypoxic stress (Fig. 6A,B). Consistently, under long-term hypoxic conditions, decreased cell viability in rPPA2 K176E rescued group compared to that of rPPA2 WT also demonstrated that rPPA2 K176E effectively counteracted PPA2-deficiency induced hypoxia resistance of tumor cells (Fig. 6C). To validate these findings in vivo, we generated a mouse xenograft liver metastasis model. Reconstitution with rPPA2 K176E notably inhibited liver metastasis compared to rPPA2 WT (Fig. 6D). This was accompanied by a decrease in the HIF-1α protein (Fig. 6E), reduced expression of its downstream target genes (Fig. 6F), and lower lactate level (Fig. 6G) in the liver metastasis samples derived from the rPPA2 K176E rescued group. These results support the notion that K176E PPA2 suppresses HIF-1α-mediated glycolytic phenotype and CRC metastasis.

Finally, to further validate the clinical relevance of PPA2 and HIF-1 signaling pathway, we performed IHC analysis on tissue sections from CRC patients (Fig. 6H). The results revealed that PPA2 expression was negatively correlated with the expression of HIF-1α and its downstream metastasis- and glycolysis-related target genes (Fig. 6I). Taken together, these results strongly suggest that PPA2 relies on K176 succinylation to inhibit HIF-1α-mediated CRC metastasis, highlighting the potential of PPA2 K176 succinylation level as a valuable biomarker for predicting mCRC risk and guiding treatment strategies.

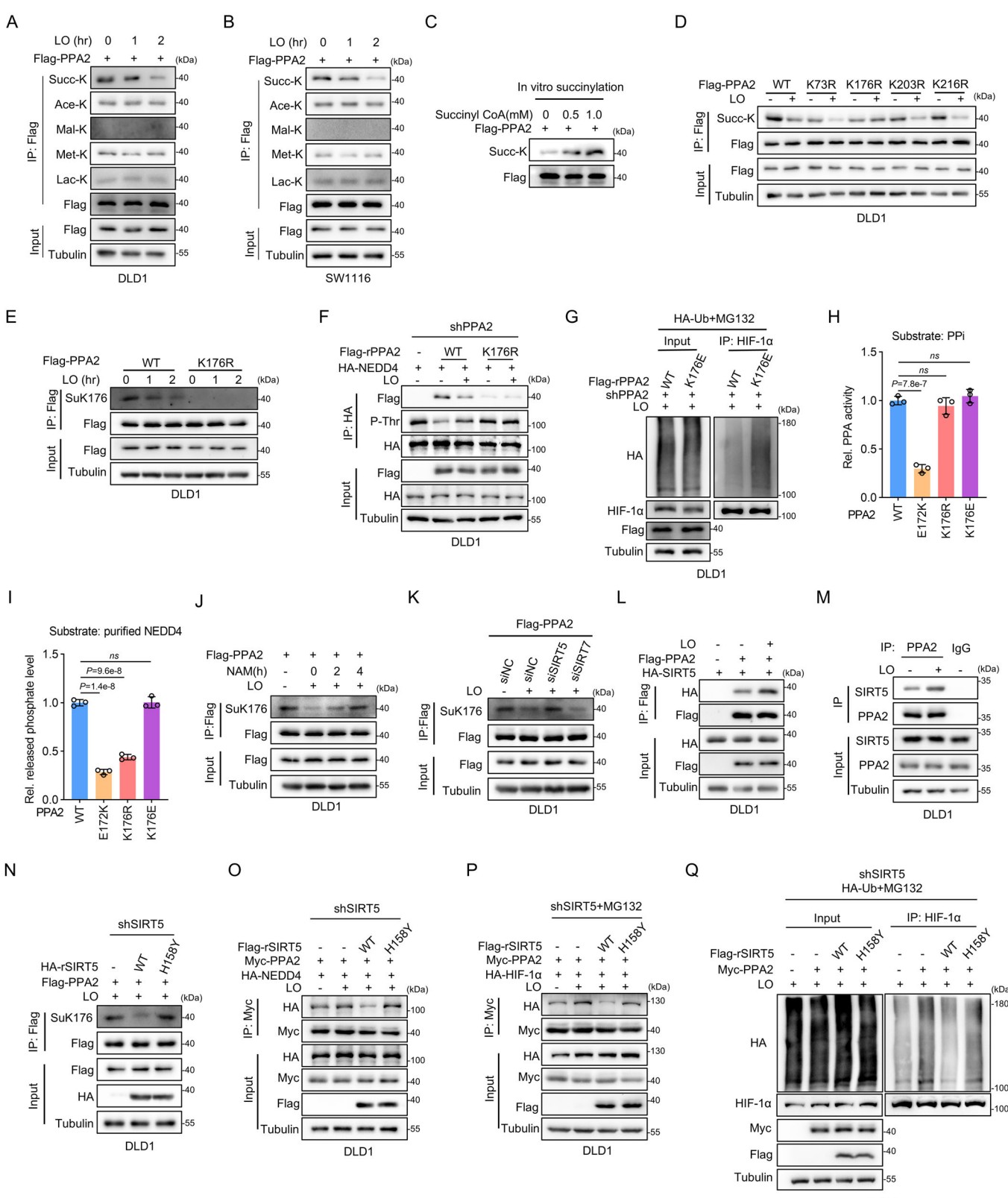

**Figure 5. SIRT5-mediated PPA2 desuccinylation at K176 impairs PPA2 function.**

(**A, B**) DLD1 and SW1116 cells overexpressing Flag-PPA2 were incubated under hypoxia for the indicated time. (**C**) Purified Flag-PPA2 proteins from HEK293T cells were incubated with different concentrations of succinyl CoA. (**D**) DLD1 cells were transfected with Flag-PPA2 WT, K73R, K176R, K203R, or K216R mutant plasmids, respectively. (**E**) DLD1 cells overexpressing Flag-PPA2 WT or K176R were incubated under hypoxia for the indicated time. (**F**) Endogenous PPA2-depleted DLD1 cells were transfected with HA-NEDD4 and EV, Flag-rPPA2 WT, or K176R mutant plasmids. (**G**) Endogenous PPA2-depleted DLD1 cells were transfected with HA-Ub and either Flag-rPPA2 WT or K176E mutant plasmids. (**H, I**) The in vitro enzymatic activity detection of purified PPA2 WT, E172K, K176R, or K176E protein from HEK293T cells. The PPi or purified NEDD4 protein from HEK293T cells were used as substrate to detect the PPA activity (**H**) and NEDD4-specific phosphatase activity (**I**). (**J**) DLD1 cells overexpressing Flag-PPA2 were treated with 5 mM NAM for the indicated time. (**K**) DLD1 cells overexpressing Flag-PPA2 were transfected with siNC, siSIRT5, or siSIRT7, respectively. (**L**) DLD1 cells were transfected with HA-SIRT5 and either EV or Flag-PPA2 plasmids. (**M**) DLD1 cells were incubated under normoxia or hypoxia and were collected for IP analysis. (**N**) Endogenous SIRT5-depleted DLD1 cells were transfected with Flag-PPA2 and either HA-rSIRT5 WT or H158Y mutant plasmids. (**O**) Endogenous SIRT5-depleted DLD1 cells were transfected with Myc-PPA2, HA-NEDD4 and either Flag-rSIRT5 WT or H158Y mutant plasmids. (**P**) Endogenous SIRT5-depleted DLD1 cells were transfected with Myc-PPA2, HA-HIF-1α and either Flag-rSIRT5 WT or H158Y mutant plasmids. (**Q**) Endogenous SIRT5-depleted DLD1 cells were transfected with HA-Ub, either Flag-rSIRT5 WT or H158Y mutant, and either EV or Myc-PPA2 plasmids. (**A–G, J–Q**) IB analyses and IP were performed using the indicated antibodies. Six hours of MG132 (5 μM) treatment was administrated as indicated. Hypoxic stress was administrated in the last 2 h before collection of cells as indicated (unless the hypoxia treatment time was specified). Data information: Data are mean ± SD from three biological replicates (**H, I**). Statistics: one-way ANOVA (**H, I**). Source data are available online for this figure.

## Discussion

In this study, we identified a critical suppressor, PPA2, in mCRC, and reported a novel mechanism regulating HIF-1α stability. PPA2 is downregulated in mCRC specimens, and low PPA2 expression correlates with a poor prognosis in CRC. Moreover, PPA2 K176 succinylation recruits and dephosphorylates E3 ligase NEDD4 to increase ubiquitin-mediated proteasomal degradation of HIF-1α. Consequently, the expression of HIF-1α target genes is inhibited by PPA2, which subsequently inhibits tumor cells' glycolytic, survival and metastatic capabilities. However, under hypoxic stress, SIRT5-mediated desuccinylation of PPA2 at K176 results in the dissociation of PPA2-NEDD4 complex, thus allowing hypoxic HIF-1α to remain stable (Fig. 7). Our findings highlight the SIRT5-PPA2-NEDD4-mediated HIF-1α stability in promoting CRC metastasis and suggest a therapeutic approach that targets HIF-1α for the treatment of mCRC.

The influential role of HIF-1α in multiple processes of cancer metastasis has been established based on accumulating evidence from preclinical studies, resulting in the recognition of HIF-1α as a promising therapeutic target for tumor metastasis. However, no efficient HIF-1α inhibitor has been successfully used in clinical trials for mCRC. A major reason is that the stability and activity of HIF-1α are subject to complex modulation. Except for the canonical VHL-mediated rapid proteasomal degradation under normoxia (Cockman et al, 2000), numerous novel regulators of HIF-1α are being identified, particularly in the context of cancer (Amelio et al, 2015; Goto et al, 2015; Kim et al, 2015; Kim et al, 2016). The existence of alternative HIF-1α regulation manners may pose challenges in achieving the expected efficacy of HIF-1α inhibitors. Therefore, enhancing our understanding of the role of HIF-1α in tumor metastasis, especially under hypoxic stress, is imperative to improve the potential clinical value of HIF-1α target. Here, we found that PPA2 inhibited HIF-1α stability via a new non-canonical ubiquitin-mediated proteasomal degradation approach. E3 ligase NEDD4 is recruited and dephosphorylated by PPA2 at T758, leading to its subsequent recognition, ubiquitination, and degradation of HIF-1α. Notably, the T758 phosphorylation site is near WW domain two, which suggests that T758 phosphorylation may interfere with the interaction of NEDD4 with HIF-1α. A similar mechanism of phosphorylation-mediated impaired substrate binding has been reported for NEDD4L, an isoform of

NEDD4 (Arroyo et al, 2011). Under normal oxygen conditions, PPA2 has minimal impact on HIF-1α stability. However, in hypoxic tumor cells, HIF-1α stability is enhanced through SIRT5-mediated desuccination of PPA2. Therefore, we propose that inhibiting SIRT5's desuccination activity on PPA2 could specifically target HIF-1α for ubiquitination and degradation in tumor cells without affecting normal cells.

Emerging evidence has shown that metabolic enzymatic activity is regulated by PTMs (Ma et al, 2024; Xu et al, 2024). Lysine succinylation is an evolutionarily conserved PTM that influences the functions of a wide range of proteins (Du et al, 2011; Kurmi et al, 2018; Song et al, 2017). Here, we discovered that succinylation of PPA2 at K176 significantly decreased under hypoxic conditions, leading to its impaired ability to bind and dephosphorylate NEDD4, which ultimately contributes to the improved stability of hypoxic HIF-1α. In addition, since no alteration in PPA activity was observed in the PPA2 K176R/E mutant, we believe that the succinylation state of PPA2 K176 specifically influences its ability to recruit and bind to a specific protein substrate. These findings suggest that PPA2 succinylation may serve as a potential biomarker for predicting the metastasis potential of CRC.

SIRT5 has recently drawn extensive attention as a highly efficient protein lysine desuccinylase and has been proven to play a vital role in the complex biological processes of tumor cells (Du et al, 2011; Park et al, 2013; Rardin et al, 2013). A recent study reported that SIRT5 desuccinylates malic enzyme 2, thus facilitating mitochondrial respiration and cancer growth in CRC (Teng et al, 2024). However, the involvement of SIRT5 in the HIF-1 signaling pathway remains largely unknown. In this study, we found that hypoxia-induced desuccinylation of PPA2 was mediated by SIRT5, as illustrated by the restored succinylation level of hypoxic PPA2 after SIRT5 knockdown and the increased interaction between SIRT5 and PPA2 under hypoxia. Moreover, the H158Y mutant of SIRT5 failed to inhibit the formation of the PPA2-NEDD4 complex and subsequent ubiquitin-mediated proteasomal degradation of HIF-1α, suggesting that the catalytic activity of SIRT5 is required for its oncogenic function in CRC. However, the mechanism by which SIRT5 responds to hypoxic stress requires further investigation.

Additionally, we identified a novel role for PPA2 in the progression of CRC, in which PPA2 deficiency in combination

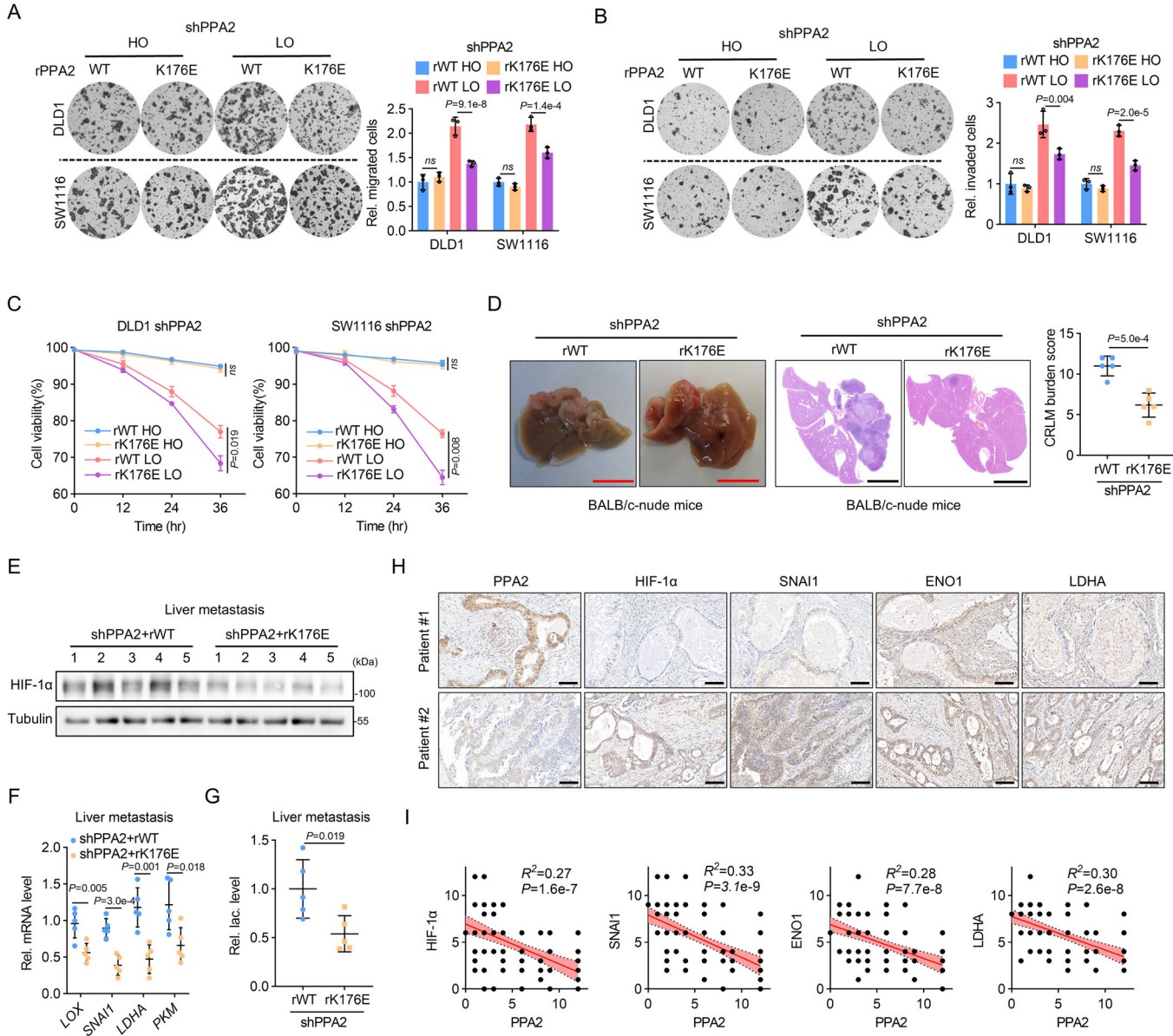

**Figure 6. PPA2 K176 succinylation is critical for mCRC suppression both in vitro and in vivo.**

(A, B) Endogenous PPA2-depleted DLD1 and SW1116 cells rescued with rPPA2 WT or K176E mutant were used for transwell migration (A) and invasion assays (B) under normoxia or hypoxia, respectively. (C) Cells in (A) were collected after incubation under hypoxia for the indicated time, and cell viability was assessed by trypan blue staining. (D–G) DLD1 cells in (A) were used for splenic injection to generate a liver metastasis model. The live image and representative HE staining of liver section were shown, and the CRLM burden was calculated in each group (D). Red scale bar, 1 cm; black scale bar, 5 mm. The liver metastasis tissues were collected to detect the HIF-1α protein level (E), the mRNA level of HIF-1α target genes (F), and the lactate level (G). (H, I) IHC analysis was performed with serial sectioned samples from 90 CRC patients, representative images of two patients were shown (H). Scale bar, 100 μm. The correlation of PPA2 level with HIF-1α, SNAI1, ENO1, and LDHA (I) was analyzed. Of note is that some of the dots on the graphs represent more than one sample (i.e., some scores overlapped). Data information: Data are mean ± SD from three biological replicates (A–C) and five biological replicates (D–G). Statistics: one-way ANOVA (A, B); two-way ANOVA (C); unpaired two-tailed student's t-test (D, F, G); Pearson's correlation analysis (I). Source data are available online for this figure.

with hypoxic stress synergistically enhanced glycolysis and migration in vitro and facilitated tumor metastasis in vivo. Interestingly, the function of PPA2 in CRC relies on its phosphatase activity, instead of its regular biological function to catalyze the intracellular hydrolysis of pyrophosphate. Here, for the first time, we investigated the effect of the phosphatase activity of PPA2 in regulating the NEDD4-mediated ubiquitination of HIF-1α. In

support of this finding, PPA family members were reported to dephosphorylate specific substrates directly (Wang et al, 2017). Moreover, we discovered that the same enzymatic site (E172) affected both the PPA and phosphatase activities of PPA2, similar to that reported for other PPA family members (Tezuka et al, 2013), which explains why the function of the PPA2 ED mutant is lost in CRC.

# Methods

### Reagents and tools table

| Reagent/Resource | Reference or Source | Identifier or Catalog Number |
|---|---|---|
| **Experimental models** | | |
| HEK293T cells (*H. sapiens*) | ATCC | CRL-3216 |
| DLD1 cells (*H. sapiens*) | ATCC | CCL-221 |
| SW1116 cells (*H. sapiens*) | ATCC | CCL-233 |
| CRC tissue samples | SYSU-FAH hospital | N/A |
| BALB/c-nude mice | Gempharmatech-GD | D000521-1 |
| **Recombinant DNA** | | |
| pCDH-SFB-hygro-PPA2-WT | This study | N/A |
| pCDH-SFB-hygro-PPA2-ED | This study | N/A |
| pCDH-HA-PPA2 | This study | N/A |
| pcDNA3.0-Myc-PPA2 | This study | N/A |
| pCDH-Flag-PPA2-WT | This study | N/A |
| pCDH-Flag-PPA2-K73R/ K176R/K203R/K216R/K176E | This study | N/A |
| pCDH-Flag-HIF-1α | This study | N/A |
| pCDH-HA-HIF-1α | This study | N/A |
| HA-Ub | This study | N/A |
| pLKO.1-shPPA2 | This study | N/A |
| pLKO.1-sgHIF-1α | This study | N/A |
| pLKO.1-shNEDD4 | This study | N/A |
| pLKO.1-shSIRT5 | This study | N/A |
| pCDH-Flag-NEDD4-FL | MiaoLing Biology | P58307 |
| pCDH-Flag-NEDD4-1 | This study | N/A |
| pCDH-Flag-NEDD4-2 | This study | N/A |
| pCDH-Flag-NEDD4-T642A/ T648A/T758A/T902A | This study | N/A |
| pCDH-HA-NEDD4 | MiaoLing Biology | P60536 |
| pCMV-HA-VHL | This study | N/A |
| pCDH-Flag-SIRT5-WT | This study | N/A |
| pCDH-HA-SIRT5-WT | This study | N/A |
| pCDH-HA-SIRT5-H158Y | This study | N/A |
| pLV2-CMV-NEDD4L-2-FLAG | MiaoLing Biology | P40389 |
| pCMV-MCS-3×Flag-WWP1 | MAILGENE | MH03569 |
| pCMV-MCS-3×Flag-WWP2 | MAILGENE | MH03570 |
| pCMV-MCS-3×Flag-Smurf1 | MAILGENE | MH04642 |
| pCMV-MCS-3×Flag-Smurf2 | MAILGENE | MH04852 |
| pCMV-MCS-3×Flag-ITCH | MAILGENE | MH01297 |
| pCMV-MCS-3×Flag-HECW2 | MAILGENE | MH04677 |
| pCMV-HECW1-3×FLAG-Neo | MiaoLing Biology | G61187 |
| pHIF1-TA-Luc | Beyotime | D4124 |
| **Antibodies** | | |
| Anti-PPA2 | Thermo Fisher Scientific | PA5-90732 |
| Anti-Alpha Tubulin | Proteintech | 11224-1-AP |
| Anti-β-Actin | Sigma-Aldrich | A1978 |
| Anti-HIF-1 Alpha | Cell Signaling Technology | 36169 |
| Anti-NEDD4 | HUABIO | HA721191 |
| Anti-Phospho-Thr | Santa Cruz | sc-5267 |
| Anti-Phospho-Ser | Santa Cruz | sc-81514 |
| Anti-Succinyl-lysine | PTMBIO | PTM-419 |
| Anti-Acetylated-Lysine | Cell Signaling Technology | 9441 |
| Anti-Malonyl-Lysine | Cell Signaling Technology | 14942 |
| Anti-Tri-Methyl-Lysine | PTMBIO | PTM-601 |
| Anti-Lactyl-Lysine | PTMBIO | PTM-1401RM |
| Anti-suK176-PPA2 | HUABIO | This study |
| Anti-Flag-tag | Proteintech | 20543 |
| Anti-HA-tag | Abmart | M20003 |
| Anti-Myc-tag | Cell Signaling Technology | 2278 |
| Anti-VHL | Cell Signaling Technology | 68547 |
| Anti-SNAI1 | Proteintech | 13099-1-AP |
| Anti-ENO1 | Proteintech | 11204-1-AP |
| Anti-LDHA | Proteintech | 19987-1-AP |
| Goat anti-Mouse IgG (H + L) Cross-Adsorbed Secondary Antibody, Alexa Fluor™ 488 | Thermo Fisher Scientific | A11001 |
| Goat anti-Rabbit IgG (H + L) Highly Cross-Adsorbed Secondary Antibody, Alexa Fluor™ Plus 594 | Thermo Fisher Scientific | A32740 |
| **Oligonucleotides and other sequence-based reagents** | | |
| qPCR primers | This study | Table EV2 |
| siRNA | This study | Table EV2 |
| shRNA | This study | Table EV2 |
| sgRNA | This study | Table EV2 |
| **Chemicals, Enzymes and other reagents** | | |
| MG-132 | Selleck | S2619 |
| NAM | Selleck | S1899 |
| Cycloheximide | Selleck | S7418 |
| IOX2 | MedChemExpress | HY-15468 |
| Lipofectamine 3000 | Thermo Fisher Scientific | L3000150 |
| Pierce IP Lysis Buffer | Thermo Fisher Scientific | 87788 |
| TRIzol reagent | Vazyme | R401-01 |
| HiScript reverse transcriptase | Vazyme | R323-01 |
| 2X Universal SYBR Green Fast qPCR Mix | Abclonal | RK21203 |
| Anti-FLAG M2 Affinity Gel | Sigma-Aldrich | A2220 |
| Protein A/G | Beyotime | P5043 |
| Polybrene | Merck Millipore | TR-1003 |
| Puromycin | MedChemExpress | HY-K1057 |

| Reagent/Resource | Reference or Source | Identifier or Catalog Number |
|---|---|---|
| 3 × Flag peptide | Sigma-Aldrich | F4799 |
| **Software** | | |
| Graphpad Prism v10 | https://www.graphpad.com/ | |
| Fiji Image J | https://imagej.net/software/fiji/downloads | |
| Zen (black version) | https://www.zeiss.com/microscopy/en/products/software/zeiss-zen.html | |
| R | https://cran.r-project.org/bin/windows/base/ | |
| NovoExpress | https://www.agilent.com.cn/zh-cn/product/research-flow-cytometry/flow-cytometry-software/novocyte-novoexpress-software-1320805 | |
| **Other** | | |
| AnnexinV-AF647/PI Apoptosis Kit | GOONIE | 100-102 |
| Dual Luciferase Reporter Assay Kit | Vazyme | DL101-01 |
| ClonExpress II One Step Cloning Kit | Vazyme | C112 |
| ChIP Assay Kit | Beyotime | P2078 |
| Malachite Green Phosphate Detection Kit | Beyotime | S0196 |
| L-Lactic Acid Colorimetric Assay Kit | Elabscience | E-BC-K044-M |
| Glucose Colorimetric Assay Kit | Elabscience | E-BC-K234-M |
| Matrigel | Corning | 354234 |
| Transwell chambers | Corning | 3422 |
| Seahorse XF Glycolysis Stress Test Kit | Agilent | 103020-100 |

## Patients and clinical samples

Tumors and adjacent colorectal tissues were obtained from CRC patients who underwent surgery or biopsy at The First Affiliated Hospital of Sun Yat-sen University. None of the patients received any chemotherapy, radiotherapy or other treatment before surgery or biopsy at the hospital. Clinicopathological information is provided in Dataset EV3. This study was approved by the hospital's ethics committee (Guangzhou, China, ethics approval: [2024]202). Written informed consent was obtained from all participants in this study. All the research was carried out in accordance with the provisions of the Declaration of Helsinki of 1975 and the Department of Health and Human Services Belmont Report.

## Cell lines, cell culture, and hypoxia treatment

DLD1, SW1116, and HEK293T cell lines were obtained from the American Type Culture Collection (ATCC). All the cells were cultured in DMEM or RPMI 1640 supplemented with 10% FBS and 100 U/mL penicillin/streptomycin. No mycoplasma contamination was detected in any cell line. The short tandem repeat (STR) profiles of cell lines are shown in Table EV1. Under normal conditions (normoxia), cells were maintained at 37 °C in a humidified incubator with 5% $CO_2$. For all experiments involving hypoxia, cells were seeded at a given density and allowed to adhere. They were then cultured in a hypoxia incubator chamber (STEMCELL, #27310), supplemented with 5% $CO_2$, 1% $O_2$, 94% $N_2$, maintaining the same temperature and humidity as in normoxia. The cell culture time was determined according to the experimental purpose.

## RNA extraction, quantitative RT–PCR (qRT-PCR)

Total RNA was extracted from cells or tissue using TRIzol reagent, and reverse transcribed using HiScript Reverse Transcriptase according to the manufacturer's instructions. qPCR amplification was performed using 2 × Universal SYBR Green Fast qPCR Mix. The qPCR primers used in this study are provided in Table EV2.

## Immunoblotting (IB) and immunoprecipitation (IP)

For IB analysis, proteins were extracted from cells or tissue by RIPA lysis buffer containing protease inhibitor and conditional phosphatase inhibitors. The protein concentration was determined using the BCA Protein Assay Kit. Protein samples were separated by SDS-PAGE, transferred onto PVDF membrane, and probed with the indicated antibodies. For IP, cells were lysed using Pierce IP lysis buffer 24 h after transfection with the indicated plasmid. Then the lysates were incubated with anti-FLAG M2 Affinity Gel or the corresponding antibodies plus protein A/G beads at 4 °C overnight, and the beads were boiled after extensive washing.

## Immunohistochemistry (IHC) analysis

Formalin-fixed, paraffin-embedded consecutive human CRC tissue sections and mouse subcutaneous tumor tissues were deparaffinized, rehydrated and heat-induced antigen retrieval in 10 mM citrate buffer (pH 6.0), then incubated with the indicated antibodies overnight at 4 °C. Subsequently, the sections were washed and incubated with secondary antibodies. Then DAB was used for rapid color development, and immunoreactivity was semi-quantitatively evaluated according to intensity and area: staining intensity was rated on a scale of 0 to 3; 0, negative; 1, weak; 2, moderate; 3, strong. The percentage of positive cells was rated on a scale of 1 to 4; 1, 0%–25%; 2, 26%–50%; 3, 51%–75%; and 4, >75%. Multiplying these numbers resulted in a final score of 0–12.

## Plasmid construction, transfection, and construction of stable cell lines

PCR-amplified human PPA2, HIF-1α, NEDD4-truncations, SIRT5 and VHL were cloned into pCDH-Flag, pCDH-hygro, pcDNA3.0-Myc, pCDH-HA vectors using the ClonExpress II One Step Cloning Kit. PPA2, NEDD4, and SIRT5 mutants were generated using the QuikChange site-directed mutagenesis kit (Stratagene). HA-Ub was obtained from The First Affiliated Hospital of Sun Yat-sen University. Transfection was performed using Lipofectamine 3000 transfection reagent according to the manufacturer's protocol.

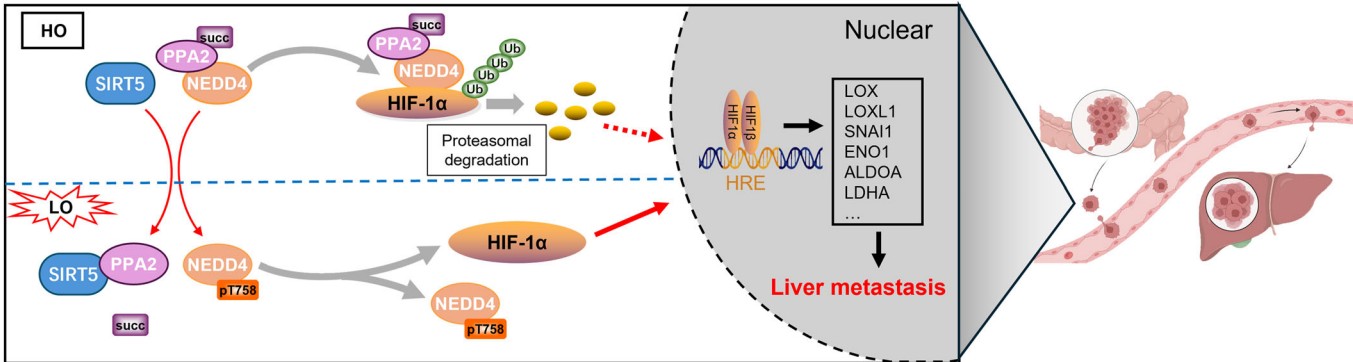

**Figure 7. Graphical model.**

Schematic diagram of succinylated PPA2 directly binding to NEDD4 to enhance its activity by dephosphorylation, thus leading to ubiquitin-medicated proteasomal degradation of HIF-1α. Impaired function of PPA2 due to hypoxia-recruited SIRT5 detaching PPA2-NEDD4 complex.

To construct knockdown cells, shRNAs targeting PPA2, NEDD4, and SIRT5 were inserted into the pLKO.1 vector. To construct knockout cells, sgRNAs targeting HIF-1α were inserted into the Lenti-CRISPR v2 vector. The psPAX2 (Addgene, #12260) and pMD2.G (Addgene, #12259) vectors were used for lentivirus production. After infection, the cells were screened with puromycin for three days. The shRNAs and sgRNAs used in this study are provided in Table EV2.

## RNA sequencing (RNA-seq) analysis

The total RNA was sequenced using an Illumina HiSeq 2500 sequencing instrument (GeneFund Biotech, Shanghai, China). To annotate genes using gene ontology (GO) terms and the KEGG pathway, ClusterProfiler v3.4.4 (R package) was used. GSEA was performed using GSEA software.

## Transwell and wound-healing assay

For the transwell assay, DLD1 and SW1116 ($1 \times 10^5$) cells were seeded in the upper compartments of transwell chambers in serum-free medium. Matrigel invasion assays were performed as described (Shaw et al, 1997). Briefly, Matrigel was diluted in cold distilled water, added to the upper well of Transwell chambers (20–30 μg/well), and dried in a sterile hood. The Matrigel was then reconstituted with medium for one hour at 37 °C before the addition of cells. The lower chambers were filled with medium containing 20% FBS. After 24–48 h of culture, the cells were fixed with 4% paraformaldehyde and stained with 0.1% crystal violet. The number of migrated and invaded cells was determined using an optical microscope.

For the wound-healing assay, DLD1 and SW1116 ($1–2 \times 10^6$) cells were seeded in 6-well plates and incubated for 12 h to allow attachment. A sterile 200 mL pipette tip was used to generate scratch wounds. The cells were cultured in media supplemented with 0.5% FBS for 24 h and imaged using an optical microscope.

## Annexin V-PI apoptosis assay

For apoptosis assays, the AnnexinV-AF647/PI Apoptosis Kit was used. Briefly, Cells were harvested and washed with cold phosphate-buffered saline (PBS), stained with AF647-conjugated

Annexin V and PI dye in binding buffer, and subjected to flow cytometric analysis within 1 h.

## Animal experiment

All animal experiments were conducted in accordance with the National Institute of Health Guide for the Care and Use of Laboratory Animals. All experimental procedures involving animals were approved by the Institutional Animal Care and Use Committee of the Sun Yat-sen University (SYSU-IACUC-2023-001489). Six-week-old female BALB/c nude mice used in this study were purchased from GemPharmatech Co., Ltd. (Guangdong, China), and maintained in a specific-pathogen-free facility. For the liver metastasis model, after anesthetization, the abdomen of each nude mouse was carefully incised in a sterile environment. $8 \times 10^5$ DLD1 cells were suspended in 100 μl PBS and injected into the subcapsular spleen, and the abdomen was closed with sutures. The mice were sacrificed six weeks after tumor cell injection. The dissected livers were photographed, fixed, and embedded in paraffin blocks for hematoxylin-eosin (HE) staining or immunohistochemistry (IHC) analysis. The CRC liver metastasis (CRLM) burden was scored according to the number and size of metastatic tumors as indicated in Table EV3.

## Detection of targeted metabolites and data analysis

All metabolites were detected using MetWare (http://www.metware.cn) on an AB Sciex QTRAP 6500 LC-MS/MS platform. Significantly regulated metabolites between the groups were determined by Variable Importance in Projection (VIP) and absolute log2FC (fold change). VIP values were extracted from the Orthogonal Partial Least Squares-Discriminant Analysis (OPLS-DA) results. The data were mean-centered before the OPLS-DA. A permutation test (200 permutations) was performed to avoid overfitting. All metabolite standards were confirmed with chemical standards; their product information and chemical abstracts service (CAS) number were listed in Appendix Table S1.

## Glucose consumption and lactate production measurement

To measure glucose consumption and extracellular lactate, $4 \times 10^5$ cells were seeded into 6-well plate and allowed to adhere. The

culture media was replaced with 2 ml complete media and was collected 12 and 24 h later for glucose and lactate content analysis using the L-Lactic Acid Colorimetric Assay Kit and the Glucose Colorimetric Assay Kit, respectively, according to the manufacturer's instructions. Glucose consumption was extrapolated by subtracting the measured glucose concentrations in the medium from the original glucose concentration (25 mM), and was normalized to total cell number.

## Extracellular acidification rate (ECAR) analysis

ECAR was measured using a Seahorse XFe96 instrument (Agilent) according to the manufacturer's instructions. Briefly, cells were seeded in assay plates the day before the analysis in a complete medium to ensure 80–90% confluence the next day. One hour before detection, cells were washed, changed to Seahorse XF Base Medium, and placed in a 37 °C $CO_2$-free incubator for 1 h. ECAR was monitored using glucose, oligomycin, and 2-deoxyglucos. The protein level in each well was determined by BCA and was applied to normalize the results.

## Dual-luciferase reporter assays

The cells were seeded in 24-well plates at a density of 70%, allowing for attachment. Thirty-six hours after transfection with pHIF1-TA-Luc and Renilla plasmids, cells were lysed for luciferase signal determination using the Dual Luciferase Reporter Assay Kit, according to the manufacturer's instructions.

## Chromatin immunoprecipitation (ChIP) assay

ChIP assays were performed using ChIP Assay Kit according to the manufacturer's instructions. Briefly, cells were cross-linked with 1% formaldehyde solution for 10 min at 37 °C and quenched with 125 mM glycine. Sheared genomic DNA ranging from 200 to 500 bp was obtained by ultrasonication and immunoprecipitated with HIF-1α or control IgG antibody overnight. Immunoprecipitated DNAs were analyzed by qPCR. The qPCR primers targeting the promoter regions of the indicated genes are listed in Table EV2.

## Liquid chromatography-mass spectrometry (LC-MS) analysis

Flag-PPA2 proteins were immunoprecipitated from DLD1 cells and separated using SDS-PAGE. LC-MS analysis of the peptides was conducted by Wininnovate Bio (Guangdong, Shenzhen) on a ThermoFisher Q Exactive mass spectrometer (ThermoFisher, USA) fitted with a Nano Flex ion source.

## Protein purification, PPA activity, and dephosphorylation assay

Flag-tagged PPA2 wild-type (WT) or mutant proteins were purified from HEK293T cells using immunoprecipitation (IP) and eluted with 3 × Flag peptide (180 μg/ml) for 8 h (h) at 4 °C. For detecting PPA activity, 100 μl reaction buffer containing 50 mM Tris, pH 8.0, 0.1 mM EGTA, 1 mM $MgCl_2$, 0.1 mM PPi and 0.5 μg recombinant PPA2 WT or mutant protein was incubated under 25 °C for 15 min. The dephosphorylation assay was performed as previously described (Jiang et al, 2023). In brief, 4 μg recombinant PPA2 WT or mutant protein was added to a phosphatase reaction buffer containing 2 μg Flag-NEDD4 protein and incubated at 37 °C for 2 h. The presence of inorganic phosphate was determined using a molybdenum dye solution.

## Generation of anti-suK176-PPA2 antibody

PPA2 K176 site-specific succinylation antibody (anti-suK176-PPA2) was generated by using a human PPA2 succinylated peptide (DVCEIGSK(su)ILS) as an antigen. The PPA2-suK176 peptide was first conjugated with keyhole limpet hemocyanin and BSA, then administered through multiple subcutaneous injections to immunize New Zealand white rabbits. Subsequently, blood was collected from the rabbits, and the presence of succinyl-specific antibodies in the serum was evaluated using enzyme-linked immunosorbent assay (ELISA) and western blot. After purifying the antibodies with excess unmodified peptide (DVCEIGSKILS), antibodies recognizing site-specific succinylation were enriched by PPA2-suK176 peptides.

## Experimental study design and statistical analysis

No sample-size calculation was performed. The sample size was determined based on our experiences and is consistent with reported literature for specific experiments. Except for the IHC scoring, the investigators were not blinded to group assignment and outcome assessment. For in vivo and in vitro assays, groups were randomly allocated. No data was excluded from the analyses. Except for RNA-seq analysis, all experiments were performed with at least three biologically independent samples or three biologically independent experiments unless stated otherwise. All statistical analyses were performed using GraphPad Prism version 8.3 (GraphPad Software, USA). A two-tailed student's t-test was used for comparisons between two groups. One-way or two-way analysis of variance (ANOVA) was used to compare multiple groups. Simple linear regression analysis was used to calculate the regression and correlation, and the log-rank test was used to compare the patient survival curves. All error bars represent mean ± SD. $P < 0.05$ was considered statistically significant.

# Data availability

Expression profile data of CRC analyzed in this study were obtained from the GEO at GSE41258 and GSE44076. Expression profile data of the pan-cancer dataset was obtained from the UCSC database (https://xenabrowser.net/). Expression profile data and available clinical information of COAD and READ cohorts were obtained from TCGA website (https://portal.gdc.cancer.gov/repository). The bulk RNA sequencing (RNA-seq) data used in this study are publicly available in GEO at GSE276272. Other data generated in this study are available in the article and appendix.

The source data of this paper are collected in the following database record: biostudies:S-SCDT-10_1038-S44318-025-00416-1.

# Peer review information

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

## Acknowledgements

This work was supported by the National Key Research And Development Plan (2022YFC3401000), National Natural Science Foundation of China (92359302, 82203152, 82303614), Guangdong Basic and Applied Basic Research Foundation (2021B1515230009, 2022A1515111079), Guangdong Provincial Key Areas R&D Programs of "Precision medicine and stem cells" (2023B1111020005), Natural Science Foundation for Outstanding Youth Team Project of Guangdong Province (2024B1515040030), and Natural Science Foundation of Guangdong Province (2024A1515013137).

## Author contributions

**Xiang Zhang**: Conceptualization; Resources; Formal analysis; Investigation; Visualization; Methodology; Writing—original draft; Writing—review and editing. **Yuqin Di**: Conceptualization; Resources; Funding acquisition; Methodology; Writing—review and editing. **Youpeng Wang**: Data curation; Software; Investigation. **Jiale Qin**: Data curation; Software. **Lvlan Ye**: Data curation. **Xiangqiong Wen**: Data curation. **Zunfu Ke**: Conceptualization; Funding acquisition. **Ziyang Wang**: Conceptualization; Supervision; Funding acquisition; Methodology; Writing—review and editing. **Weiling He**: Conceptualization; Supervision; Funding acquisition.

Source data underlying figure panels in this paper may have individual authorship assigned. Where available, figure panel/source data authorship is listed in the following database record: biostudies:S-SCDT-10_1038-S44318-025-00416-1.

## Disclosure and competing interests statement

The authors declare no competing interests.

# Expanded View Figures

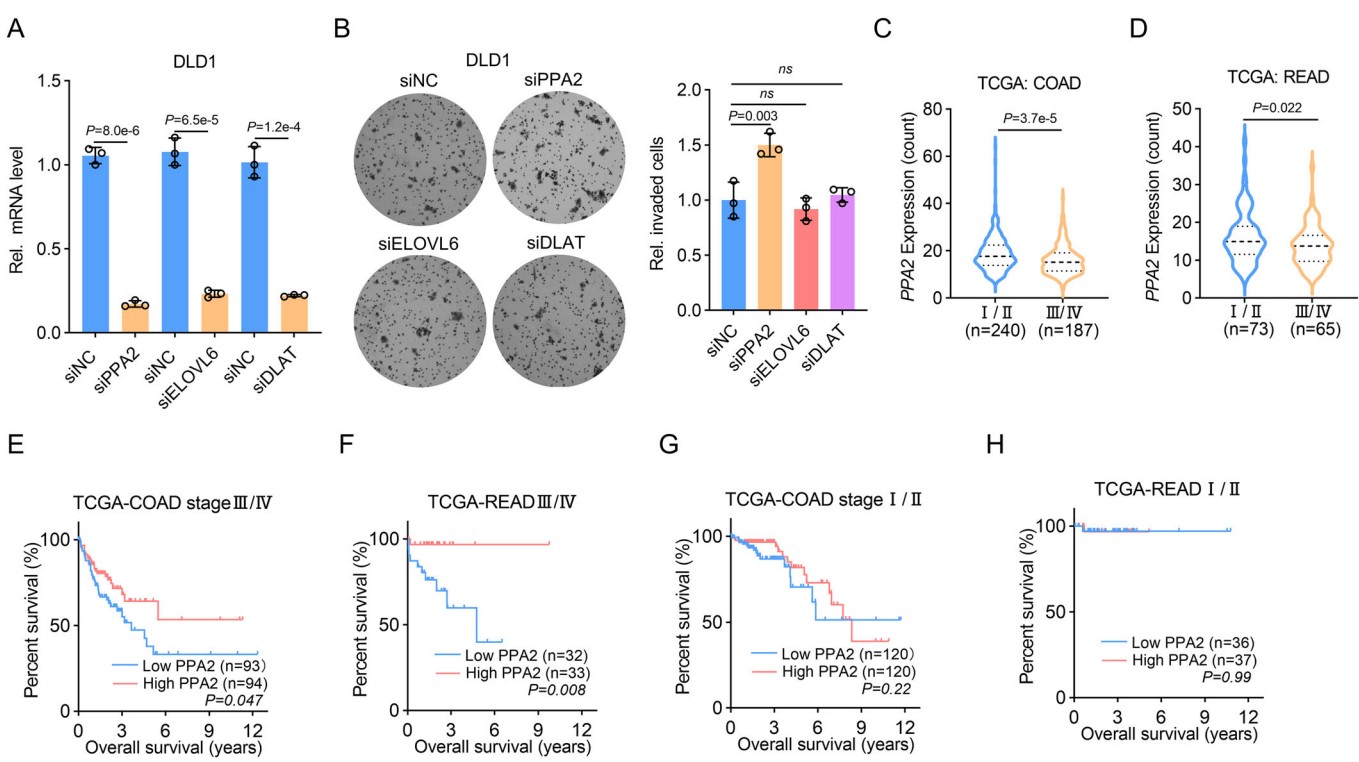

**Figure EV1. PPA2 downregulation correlates with increased CRC metastasis and poor prognosis.**

(A) qRT-PCR validated the efficiency of PPA2, ELOVL6 and DLAT knockdown in DLD1 cells. (B) Cells in (A) were used for transwell invasion assay. (C, D) TCGA RNA-seq data of patients with COAD (C) and READ (D) were analyzed. *PPA2* mRNA levels were compared between different stages. (E–H) Kaplan–Meier analysis of survival durations based on the PPA2 expression status at advanced stages (E, F) or early stages (G, H) from TCGA-COAD (E, G) or TCGA-READ (F, H) datasets. Data information: Data are mean ± SD from three biological replicates (A, B). Statistics: unpaired two-tailed student's t-test (A, C, D); one-way ANOVA (B); log-rank test (E–H). Source data are available online for this figure.

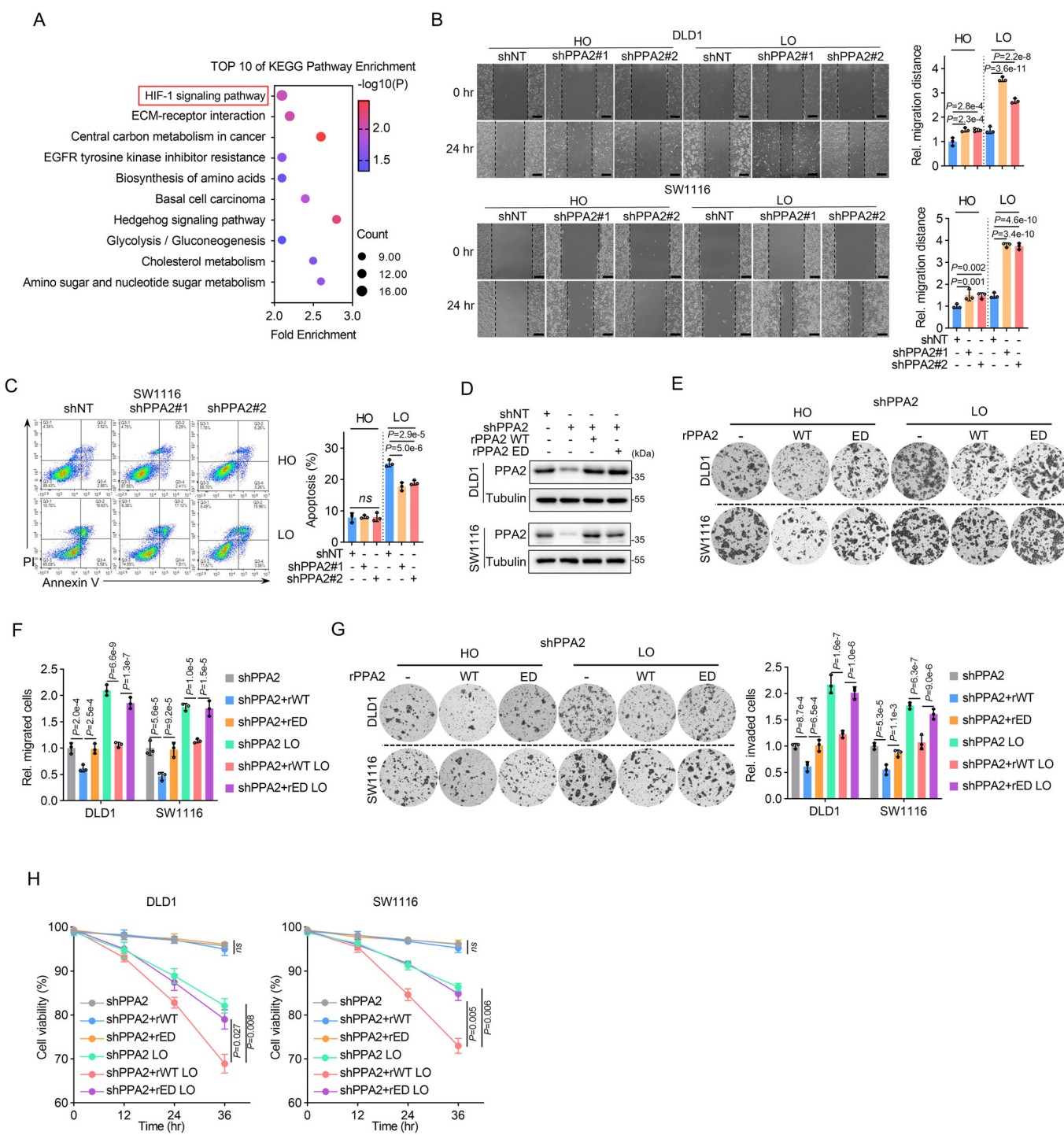

**Figure EV2. PPA2 inhibits tumor cell migration and hypoxia resistance in an enzymatic activity-dependent manner.**

(A) KEGG analysis of pathways upregulated in PPA2-depleted compared to control DLD1 cells. (B) Control or PPA2-depleted DLD1 and SW1116 cells were used for wound-healing assay under normoxia or hypoxia, respectively. Scale bar, 200 μm. (C) Control or PPA2-depleted SW1116 cells were collected after incubation under normoxia or hypoxia, respectively; PI/Annexin V double staining was then used to measure the percentage of apoptotic cells via flow cytometry. (D) IB analysis in control or endogenous PPA2-depleted DLD1 and SW1116 cells reconstituted with rPPA2 WT or ED mutant. (E–G) Cells in (D) were used for transwell migration (E, F) and invasion (G) assay under normoxia or hypoxia, respectively. (H) Cells in (D) were collected after incubation under normoxia or hypoxia for the indicated time, and cell viability was assessed by trypan blue staining. Data information: Data are mean ± SD from three biological replicates (B, C, F–H). Statistics: one-way ANOVA (B, C, F, G); two-way ANOVA (H). Source data are available online for this figure.

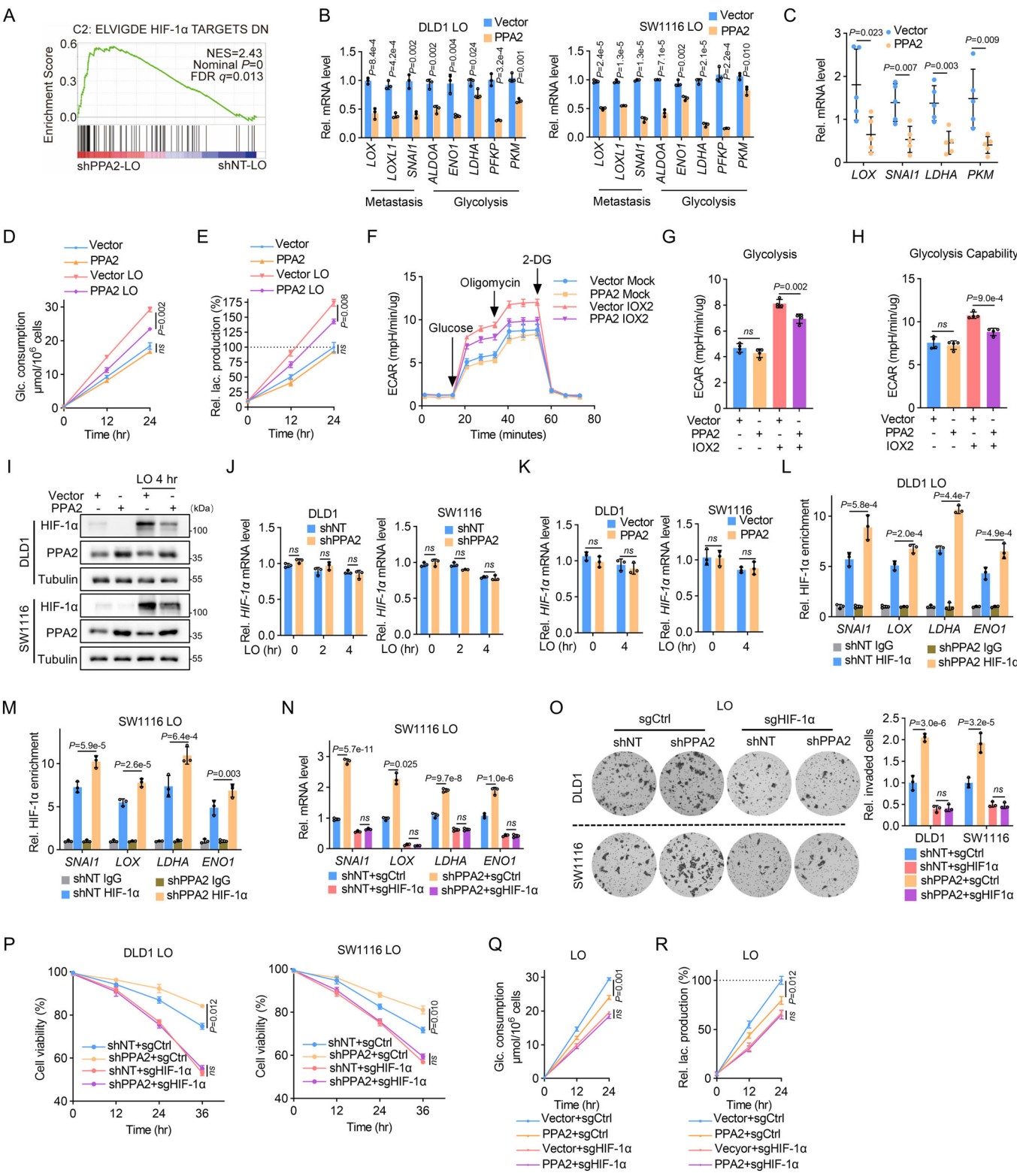

◀   **Figure EV3.   PPA2 inhibits glycolysis-mediated CRC metastasis by facilitating the degradation of HIF-1α.**

(A) GSEA analysis of HIF-1α targets upregulated in PPA2-depleted compared to control DLD1 cells after incubation under hypoxia for 12 h. NES, normalized enriched score. FDR, False discovery rate. (B) DLD1 and SW1116 cells overexpressing vector or PPA2 were collected after incubation under hypoxia for 12 h, the mRNA level of HIF-1α target genes were analyzed via qRT-PCR. (C) The mRNA level of the HIF-1α target genes was analyzed via qRT-PCR in liver metastasis tissues derived from splenic injection using PPA2-overexpressed DLD1 cells. (D, E) The media of DLD1 cells overexpressing vector or PPA2 were collected after incubation under normoxia or hypoxia for the indicated time to detect the glucose (D) and lactate levels (E). The lactate level of vector group at 24 h was used as the standard for data normalization in (E). (F–H) DLD1 cells overexpressing vector or PPA2 were used for ECAR experiment with or without IOX2. (I) Control or PPA2-overexpressed DLD1 and SW1116 cells were collected after incubation under normoxia or hypoxia induction for 4 h. IB analysis was performed using the indicated antibodies. (J, K) PPA2-depleted (J) or PPA2-overexpressed (K) DLD1 and SW1116 cells were collected after incubation under normoxia or hypoxia for the indicated time to detect the mRNA level of *HIF-1α* via qRT-PCR. (L, M) Control or PPA2-depleted DLD1 and SW1116 cells were collected after incubation under hypoxia for 6 h. ChIP assays were performed with the indicated antibodies, and analyzed via qPCR with primers targeting the promoter region of the indicated genes. (N) Control or PPA2-depleted SW1116 cells treated with or without sgRNA targeting HIF-1α were collected after incubation under hypoxia for 12 h to detect the mRNA level of HIF-1α target genes via qRT-PCR. (O) Control or PPA2-depleted DLD1 and SW1116 cells treated with or without sgRNA targeting HIF-1α were used for transwell invasion assays under hypoxia. (P) Cells in (O) were collected after incubation under hypoxia for the indicated time, and cell viability was assessed by trypan blue staining. (Q, R) The media of DLD1 cells in (O) were collected after incubation under hypoxia for the indicated time to detect the glucose (Q) and lactate levels (R). The lactate level of vector+sgCtrl group at 24 h was used as the standard for data normalization in (R). Data information: *n* = two biological replicates for each group (A). Data are mean ± SD from three biological replicates (B, D, E, J–R), five biological replicates (C) and four biological replicates (F–H). Statistics: permutation test on enrichment score was performed following Subramanian algorithm (A); unpaired two-tailed student's t-test (B, C, J, K); one-way ANOVA (G, H, L–O); two-way ANOVA (D, E, P-R). Source data are available online for this figure.

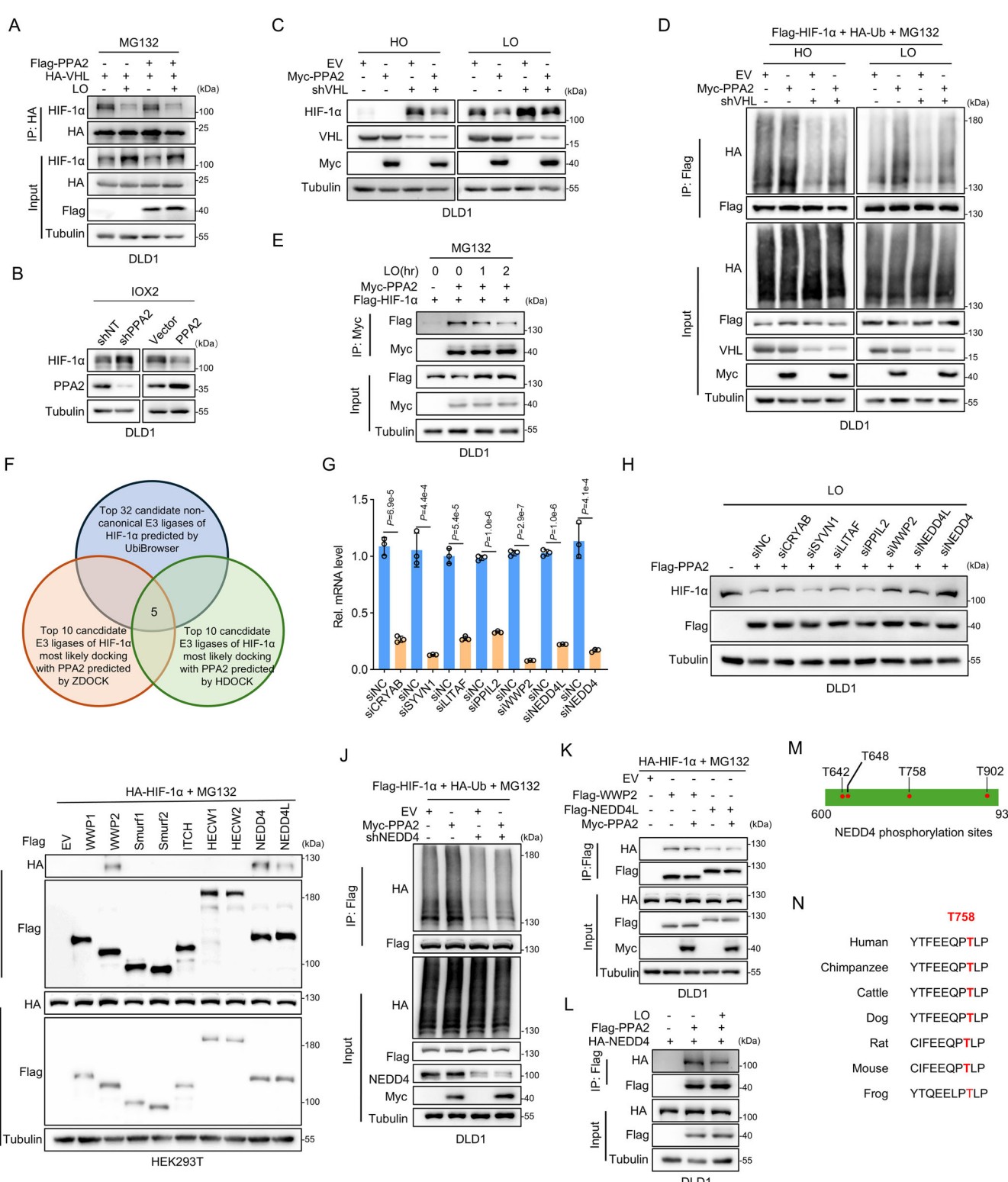

◀

**Figure EV4. PPA2 recruits and activates NEDD4 for ubiquitin-mediated proteasomal degradation of HIF-1α.**

(**A**) DLD1 cells transfected with HA-VHL and either empty vector (EV) or Flag-PPA2 plasmids. (**B**) The indicated DLD1 cells were treated with 40 μM IOX2 for 2 h. (**C**) DLD1 cells stably expressing shNT or shVHL were transfected with either EV or Myc-PPA2 plasmids. (**D**) DLD1 cells stably expressing shNT or shVHL were transfected with Flag-HIF-1α, HA-Ub and either EV or Myc-PPA2 plasmids. (**E**) DLD1 cells transfected with Flag-HIF-1α and either EV or Myc-PPA2 plasmids were incubated under hypoxia for the indicated time. (**F**) The candidate E3 ligases predicted by the UbiBrowser database were subjected to molecular docking analyses with PPA2 via ZDOCK and HDOCK, and the top 5 were sorted by the docking score. (**G**) PPA2-overexpressed DLD1 cells were transfected with siNC, siCRYAB, siSYVN1, siLITAF, siPPIL2, siWWP2, siNEDD4L, or siNEDD4 respectively, and the knockdown efficiencies were validated by qRT-PCR. (**H**) DLD1 cells overexpressing vector or Flag-PPA2 were transfected with siNC, siCRYAB, siSYVN1, siLITAF, siPPIL2, siWWP2, siNEDD4L, or siNEDD4 respectively. (**I**) DLD1 cells were transfected with HA-HIF-1α and EV, Flag-WWP1, WWP2, Smurf1, Smurf2, ITCH, HECW1, HECW2, NEDD4, or NEDD4L plasmids, respectively. (**J**) DLD1 cells stably expressing shNT or shNEDD4 were transfected with Flag-HIF-1α, HA-Ub and either EV or Myc-PPA2 plasmids. (**K**) DLD1 cells transfected with HA-HIF-1α and along with either EV, Flag-WWP2, Flag-NEDD4L, or Myc-PPA2 plasmids. (**L**) DLD1 cells were transfected with HA-NEDD4 and either EV or Flag-PPA2 plasmids. (**M**) Schematic diagram of potential threonine phosphorylation residues within 600-930AA of NEDD4 from PhosphoSitePlus. (**N**) Sequence alignment of NEDD4 T758 peptides in the indicated species. (**A–E, H–L**) IB analyses and IP were performed using the indicated antibodies. Six hours of MG132 (5 μM) treatment was administrated as indicated. Hypoxic stress was administrated in the last 2 h before the collection of cells as indicated (unless the hypoxia treatment time was specified). Data information: Data are mean ± SD from three biological replicates (**G**). Statistics: unpaired two-tailed student's t-test (**G**). Source data are available online for this figure.

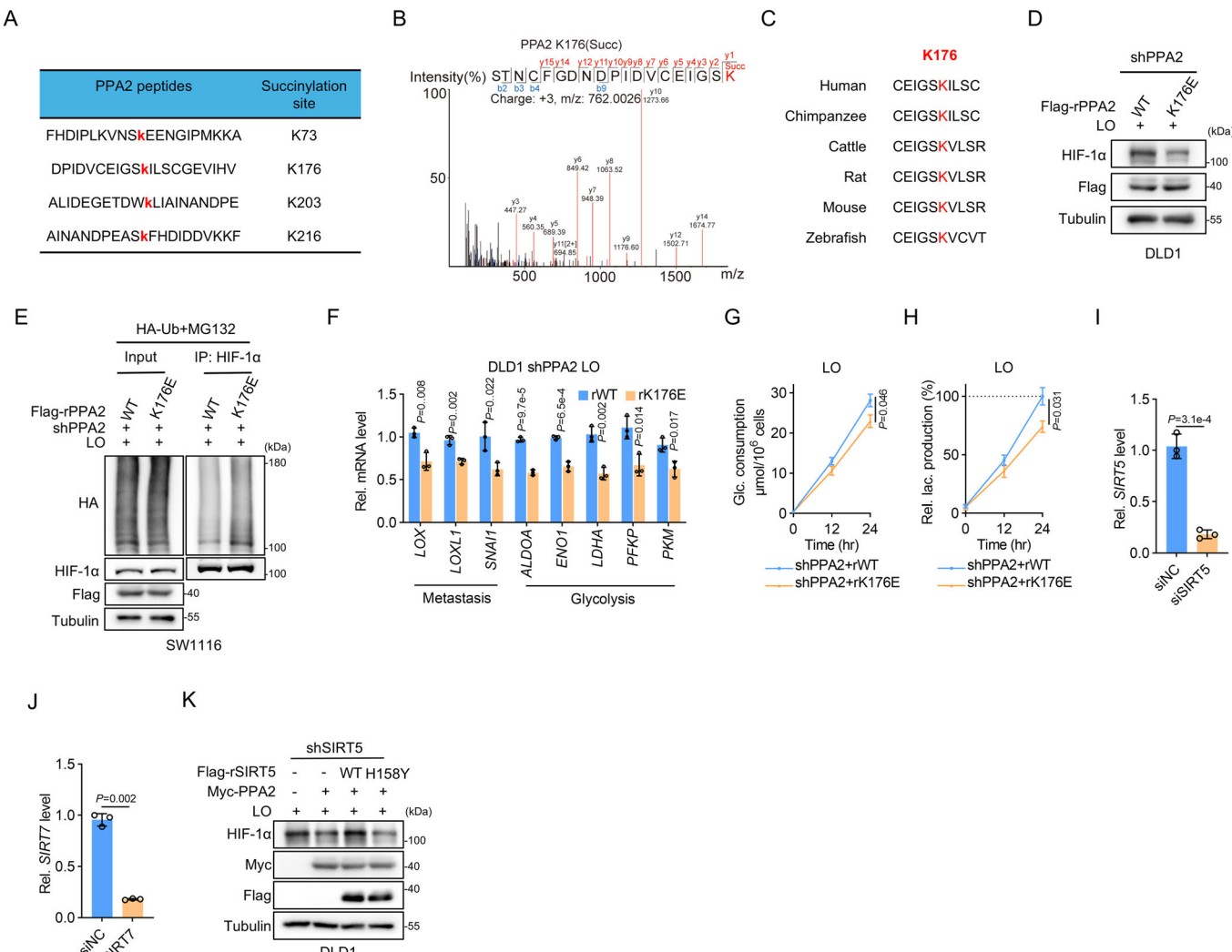

**Figure EV5. SIRT5-mediated PPA2 desuccinylation at K176 impairs PPA2 function.**

(A) Hypoxia induced desuccinylation of PPA2 was detected by LC-MS analysis at four lysine residues. (B) The LC-MS spectrum of hypoxia-induced modified K176 of PPA2 is shown. (C) Sequence alignment of PPA2 K176 peptides in the indicated species. (D) Endogenous PPA2-depleted DLD1 cells were transfected with Flag-rPPA2 WT or K176E mutant plasmids. (E) Endogenous PPA2-depleted SW1116 cells were transfected with HA-Ub and either Flag-rPPA2 WT or K176E mutant plasmids. (F) Endogenous PPA2-depleted DLD1 cells transfected with Flag-rPPA2 WT or K176E plasmids were incubated under hypoxia for 12 h, and were collected to detect the mRNA level of HIF-1α target genes. (G, H) The media of DLD1 cells in (F) were collected after incubation under hypoxia for the indicated time to detect the glucose (G) and lactate levels (H). The lactate level of shPPA2+rWT group at 24 h was used as the standard for data normalization in (H). (I, J) The knockdown efficiency of siSIRT5 (I) and siSIRT7 (J) in DLD1 cells was confirmed by qRT-PCR. (K) Endogenous SIRT5-depleted DLD1 cells were transfected with Flag-rSIRT5 WT or H158Y mutant and either EV or Myc-PPA2 plasmids. (D, E, K) IB analysis and IP were performed using the indicated antibodies. Six hours of MG132 (5 μM) treatment was administrated as indicated. Hypoxic stress was administrated in the last 2 h before collection of cells as indicated. Data information: Data are mean ± SD from three biological replicates (F–J). Statistics: unpaired two-tailed student's t-test (F, I, J); two-way ANOVA (G, H). Source data are available online for this figure.

