## [Peer Review File · The EMBO Journal]

SIRT5-mediated desuccinylation of PPA2 enhances HIF-1 α -dependent adaptation to hypoxic stress and colorectal cancer metastasis

Xiang Zhang, Yuqin Di, Youpeng Wang, Jiale Qin, Lvlan Ye, Xiangqiong Wen, Zunfu Ke, Ziyang Wang, and Weiling He

Corresponding author(s): Ziyang Wang (wangzy256@mail.sysu.edu.cn) , Zunfu Ke (kezunfu@mail.sysu.edu.cn), Weiling He (wlhe@xah.xmu.edu.cn)

Review Timeline:

Submission Date:	8th Oct 24
Editorial Decision:	13th Nov 24
Revision Received:	28th Jan 25
Editorial Decision:	20th Feb 25
Revision Received:	25th Feb 25
Accepted:	27th Feb 25

Editor: Daniel Klimmeck

Transaction Report:

Dear Dr Wang,

Thank you again for the submission of your manuscript (EMBOJ-2024-119243) to The EMBO Journal. Please accept my apologies for getting back to you with unusual delay due to protracted referee input and detailed discussion in the editorial team. As mentioned earlier, your study was assessed by two reviewers with expertise in cancer biology and hypoxia signaling, whose comments are enclosed below.

As you will see from the experts' reports, the referees acknowledge the analysis and potential interest of your results. However, they also express important concerns regarding completeness and robustness of the findings, which need to be addressed thoroughly to make them supportive of publication in the EMBO Journal. The reviewers also raise issues related to the data presentation, additional controls and improved methods annotation required, statistics applied and overall discussion of related literature, that would need to be conclusively addressed to achieve the level of robustness and clarity needed for The EMBO Journal.

Given the overall interest stated and broader angle of your findings, we are able to invite you to revise your manuscript experimentally to address the referees' comments. I need to stress though that we do require strong support from the referees on a revised version of the study in order to move on to publication of the work.

Please feel free to contact me if you have any questions or need further input on the referee comments.

When submitting your revised manuscript, please carefully review the instructions below.

Please feel free to approach me any time should you have additional questions related to this.

Thank you for the opportunity to consider your work for publication.

I look forward to your revision.

Kind regards,

Daniel Klimmeck

Daniel Klimmeck, PhD
Senior Editor
The EMBO Journal

Instruction for the preparation of your revised manuscript:

- 1) a .docx formatted version of the manuscript text (including legends for main figures, EV figures and tables). Please make sure that the changes are highlighted to be clearly visible.
- 2) individual production quality figure files as .eps, .tif, .jpg (one file per figure).
- 3) a .docx formatted letter INCLUDING the reviewers' reports and your detailed point-by-point response to their comments. As part of the EMBO Press transparent editorial process, the point-by-point response is part of the Review Process File (RPF), which will be published alongside your paper.
- 4) a complete author checklist, which you can download from our author guidelines ([https://wol-prod-cdn.literatumonline.com/pb-assets/embo-site/Author Checklist%20-%20EMBO%20J-1561436015657.xlsx](https://wol-prod-cdn.literatumonline.com/pb-assets/embo-site/Author%20Checklist%20-%20EMBO%20J-1561436015657.xlsx)). Please insert information in the checklist that is also reflected in the manuscript. The completed author checklist will also be part of the RPF.
- 5) Please note that all corresponding authors are required to supply an ORCID ID for their name upon submission of a revised

manuscript.

6) It is mandatory to include a 'Data Availability' section after the Materials and Methods. Before submitting your revision, primary datasets produced in this study need to be deposited in an appropriate public database, and the accession numbers and database listed under 'Data Availability'. Please remember to provide a reviewer password if the datasets are not yet public (see <https://www.embopress.org/page/journal/14602075/authorguide#datadeposition>).

7) Our journal encourages inclusion of *data citations in the reference list* to directly cite datasets that were re-used and obtained from public databases. Data citations in the article text are distinct from normal bibliographical citations and should directly link to the database records from which the data can be accessed. In the main text, data citations are formatted as follows: "Data ref: Smith et al, 2001" or "Data ref: NCBI Sequence Read Archive PRJNA342805, 2017". In the Reference list, data citations must be labeled with "[DATASET]". A data reference must provide the database name, accession number/identifiers and a resolvable link to the landing page from which the data can be accessed at the end of the reference. Further instructions are available at .

8) At EMBO Press we ask authors to provide source data for the main and EV figures. Our source data coordinator will contact you to discuss which figure panels we would need source data for and will also provide you with helpful tips on how to upload and organize the files.

Numerical data can be provided as individual .xls or .csv files (including a tab describing the data). For 'blots' or microscopy, uncropped images should be submitted (using a zip archive or a single pdf per main figure if multiple images need to be supplied for one panel). Additional information on source data and instruction on how to label the files are available at .

9) We replaced Supplementary Information with Expanded View (EV) Figures and Tables that are collapsible/expandable online (see examples in <https://www.embopress.org/doi/10.15252/embj.201695874>). A maximum of 5 EV Figures can be typeset. EV Figures should be cited as 'Figure EV1, Figure EV2' etc. in the text and their respective legends should be included in the main text after the legends of regular figures.

11) For data quantification: please specify the name of the statistical test used to generate error bars and P values, the number (n) of independent experiments (specify technical or biological replicates) underlying each data point and the test used to calculate p-values in each figure legend. The figure legends should contain a basic description of n, P and the test applied. Graphs must include a description of the bars and the error bars (s.d., s.e.m.).

We realize that it is difficult to revise to a specific deadline. In the interest of protecting the conceptual advance provided by the work, we recommend a revision within 3 months (11th Feb 2025). Please discuss the revision progress ahead of this time with the editor if you require more time to complete the revisions.

Referee #1:

The manuscript from Zhang and colleagues describes a novel mode of regulation of HIF1a stability through the activity of PPA2 and NEDD4. However, NEDD4 is not a single protein, but describes in mammals a family of proteins, of which in humans there are thought to be nine members. Some work has previously shown an interaction between a member in this family; NEDD4L (also known as NEDD4-2) and HIF1a (Liu et al. Cell Reports 2023), which should be referenced in the manuscript. Therefore it is critical for the authors to define further which family member they are referring to, which they are manipulating in their experiments, and if multiple family members bind and regulate HIF1a, whether this interaction is influenced by PPA2 activity in all cases.

Other major comments:

- It isn't clear exactly how hypoxia is induced and maintained - the description in the materials and methods only says what the oxygen tension was (1%). What is important is to state the type of glovebox, workstation or incubator used, and if not in a specific incubator or workstation, how the precise gas mixture was maintained at the experimental settings (i.e. 1% oxygen).
- The authors rely significantly on a reference from Loo et al. (2015) to suggest that CRC cells disseminating to the liver through the portal circulation experience hypoxic conditions that can result in abnormal cell death (Lines 174-176). The reference in question does not support this statement. They suggest that the portal circulation is relatively hypoxemic (it maintains >60% hemoglobin saturation), and that they hypothesized that this might result in acute hypoxic stress. Their data show that CRC cells exhibit hypoxia early in colonization, not that the most significant effect is within the hepatic portal vein, in which they are unlikely to spend much time. Please could the authors reflect this in the text where they cite this reference throughout.
- Supplementary Figure 2E - need to see the siPPA2 only control alongside the +WT and +ED so that the effect of recovering WT PPA2 expression can be observed.
- In Figure 2M&N, the authors present clear data that knockdown of PPA2 increases the tumor burden in the liver. However, the re-expression study (N) shows only the wild-type and not the ED version. This is surprising, given that the authors up to this point used both in parallel in the in vitro studies. It would add significant weight to the study if the authors had these data.
- Unclear what the difference is between Figure 1B and Supplementary Figure 1B - how exactly is invasion being differentiated from migration given that a transwell system is being used for both?
- The metabolomic studies outlined in Figure 3 are interesting, but are not sufficient to conclude enhanced glycolysis. Changes in glycolytic intermediates can be independent of the flux through the pathway - they could increase with a block in glycolysis just as easily. To show enhanced glucose uptake or lactate synthesis, intracellular levels of glucose are also not sufficient. Instead, measurements on the media showing whether glucose is being depleted at a faster rate, and whether lactate is being exported at an increased rate will provide those data.
- In Figure 3H, CoCl₂ cannot be used as a hypoxia mimetic for metabolic experiments - high doses of cobalt ions can block other divalent ion transporters, and in doing so can alter metabolism. A small molecule PHD enzyme inhibitor (not the aKG-like DMOG) would be more suitable for this study.
- In general, it would be better if the authors could use a more recent, more specific PHD inhibitor in their studies, such as IOX2.
- The data in Supplementary Figure 4B show that shPPA2 leads to increased stabilization of HIF1a in the presence of CoCl₂. This does not necessarily rule out an activity for pVHL at this point - the proof for this would be to use VHL-deficient cells and overexpress PPA2 to show that this decreases its stability in this system. I believe that this experiment would significantly strengthen the manuscript
- From Line 341, the authors interrogate lysine modifications. It is a considerable leap to move straight to succinylation as the potential target modification, when there are a significant number of potential candidates, as tested in the blot in Figure 5A. Could the authors also refer to testing the other modifications shown?
- Using a lysine to arginine point mutation to mimic non-succinylated lysine is not a common approach - could the authors reference this having been used before and demonstrated to work in this way. The same point is relevant for the use of a point

mutation to glutamate to mimic succinylation - a reference is required demonstrating this point.

- The evidence for PPA2 K176 residue being involved in the desuccinylation in hypoxia is shown in the blot in Figure 5D. However, this level of evidence could mean that binding of something to this residue is required for desuccinylation of another residue, as much as it is the residue that is directly desuccinylated in hypoxia. The use of point mutants is not sufficient either, as this could also disrupt a putative binding site. Further direct proof is required to conclude that this residue is specifically desuccinylated in hypoxia

- The data in Figure 6I is not sufficient to conclude that HIF1a protein and PPA2 protein are negatively correlated. The R2 values are very small, suggesting little correlation, despite a mathematical significant level being reached.

Minor

- Lines 149-150; the evidence at this point is correlative, and nothing shows a direct role to here. I think the sentence should be changed to reflect that reduced expression of PPA2 is correlated with disease progression.

- Similarly, the role that the authors are trying to elicit using their cell culture models is not PPA2 in progression, but in cell phenotype (line 163).

- Methods for the metabolomics study does not reveal whether metabolite IDs were confirmed with chemical standards - can this be clarified in the methods section.

Referee #2:

Summary:

The manuscript by Zhang, Di, and Wang et al., investigated the role of PPA2 and hypoxia in relation to colorectal cancer metastasis. The authors found that during periods of low oxygen PPA2 decreases HIF-1a stability via NEDD4. Furthermore, the authors showed that Sirt5 promotes dissociation of PPA2 and NEDD4 via succinylation and increases HIF-1a stability. Overall, the conclusions of this paper are justified by the data. I would recommend this manuscript for publication with some minor additions.

Major Comments:

1. Can the authors show if the opposite metabolic alterations happen in PPA2 overexpression and sgHIF1a cells?
2. How does OE of PPA2 influence downstream HIF target genes?
3. Does PPA2 expression have an influence on primary tumor growth or only promotion of metastasis?
4. In general, does CRC actually experience hypoxia?

Minor Comments:

1. 3R color legend order is different from how data presented in figure

Responses to Reviewers' Comments

We deeply appreciate the reviewers for their valuable and constructive suggestions, as well as their support in enhancing our manuscript. As suggested, we have performed additional experiments to validate our findings and refined several descriptions in the manuscript. All the modifications made in the manuscript are highlighted in yellow.

Our point-by-point responses are provided below.

Reviewer comments:

Referee #1:

The manuscript from Zhang and colleagues describes a novel mode of regulation of HIF1a stability through the activity of PPA2 and NEDD4. However, NEDD4 is not a single protein, but describes in mammals a family of proteins, of which in humans there are thought to be nine members. Some work has previously shown an interaction between a member in this family; NEDD4L (also known as NEDD4-2) and HIF1a (Liu et al. Cell Reports 2023), which should be referenced in the manuscript. Therefore it is critical for the authors to define further which family member they are referring to, which they are manipulating in their experiments, and if multiple family members bind and regulate HIF1a, whether this interaction is influenced by PPA2 activity in all cases.

Response: We thank you for the insightful suggestions that have strengthened our study. To address your concerns, we reviewed some related research, performed additional experiments, and revised the result section of the manuscript, as quoted below:

"NEDD4 is one member of the NEDD4 subfamily, which consists of nine members in humans, each having overlapped yet distinct repertoires of binding partners/substrates (PMID: 26116757, Biochimica et biophysica acta, 2015). To determine whether multiple NEDD4 family members can bind to HIF-1 α , we performed exogenous co-IP in DLD1 cells. The results revealed that only WWP2, NEDD4 and NEDD4L notably associated with HIF-1 α , while the other six members did not show any interaction (Fig. EV4I). Additionally, knockdowns of WWP2 or NEDD4L did not fully rescue the PPA2-induced decrease in hypoxic HIF-1 α protein levels as effectively as NEDD4 did; instead, they resulted in only a moderate increase in HIF-1 α (Fig. EV4G, H), corroborating previous findings by Liu *et al.* (PMID: 37494179, Cell reports, 2023). Collectively, these results suggest that PPA2 relies on NEDD4 for the degradation of HIF-1 α . (Page 11, new line 304-314) ...Co-IP results revealed that the interaction between NEDD4 and HIF-1 α was robust under normoxia but considerably weakened under hypoxic conditions (Fig. 4D). Notably, PPA2 strengthened this interaction under both normoxic and hypoxic conditions (Figure

4D); however, it did not influence the association between HIF-1 α and WWP2 or NEDD4L (Fig. EV4K), further excluding the possibility that WWP2 or NEDD4L is involved in the regulation of HIF-1 α by PPA2. (Page 12, new line 321-324)”

To clearly address your questions, we have cited the literature you mentioned (as indicated in bold in the previous text). Additionally, it should be stated that, in our research, we study NEDD4, rather than the other members of the NEDD4 family. We would like to thank you again for your valuable comments, which have provided stronger evidence for our research.

Other major comments:

- It isn't clear exactly how hypoxia is induced and maintained - the description in the materials and methods only says what the oxygen tension was (1%). What is important is to state the type of glovebox, workstation or incubator used, and if not in a specific incubator or workstation, how the precise gas mixture was maintained at the experimental settings (i.e. 1% oxygen).

Response: Thank you for this comment. We have added the statement in the material and method section (Page 20, new line 561-565), as quoted below:

“For all experiments involving hypoxia, cells were seeded at a given density and allowed to adhere. They were then cultured in a hypoxia incubator chamber (STEMCELL, #27310), supplemented with 5% CO₂, 1% O₂, 94% N₂, maintaining the same temperature and humidity as in normoxia.”

- The authors rely significantly on a reference from Loo et al. (2015) to suggest that CRC cells disseminating to the liver through the portal circulation experience hypoxic conditions that can result in abnormal cell death (Lines 174-176). The reference in question does not support this statement. They suggest that the portal circulation is relatively hypoxemic (it maintains >60% hemoglobin saturation), and that they hypothesized that this might result in acute hypoxic stress. Their data show that CRC cells exhibit hypoxia early in colonization, not that the most significant effect is within the hepatic portal vein, in which they are unlikely to spend much time. Please could the authors reflect this in the text where they cite this reference throughout.

Response: Thank you for this comment. We apologize for our insufficient understanding of this literature and any potential misunderstandings it may have caused.

We have revised the statement in the results section (Page 7, new line 176-179), as quoted below:

“Given that CRC cells often encounter hypoxic conditions during the metastatic process, such as acute hypoxia that may occur when passing through the hypoxemic portal vein and during early colonization of the hypoxic liver microenvironment,

which results in abnormal death in tumor cells in this process (Loo *et al.*, 2015).”

- Supplementary Figure 2E - need to see the siPPA2 only control alongside the +WT and +ED so that the effect of recovering WT PPA2 expression can be observed.

Response: Thank you for this comment. To present the effect of enzymatically dead PPA2 mutant (ED) more concisely, we initially did not include the results of the shPPA2 group as a control, although it was part of the original experimental design. We have since updated the results in the new Fig. EV2E-H to include the shPPA2 group, which consistently demonstrated that, compared to WT, the reconstitution of rPPA2 ED failed to counteract the effects of shPPA2. We appreciate your suggestion, as it has helped us better present our results.

- In Figure 2M&N, the authors present clear data that knockdown of PPA2 increases the tumor burden in the liver. However, the re-expression study (N) shows only the wild-type and not the ED version. This is surprising, given that the authors up to this point used both in parallel in the *in vitro* studies. It would add significant weight to the study if the authors had these data.

Response: Thank you for this comment. We conducted an *in vivo* liver metastasis study using DLD1 cells that overexpressed either PPA2 WT or ED. Our results indicated that the CRC liver metastasis (CRLM) burden was significantly inhibited by the overexpression of PPA2 WT, but not by the ED mutant, compared to control groups (new Fig. 2N). This consistently demonstrated that the *in vivo* anti-metastatic effects of PPA2 also depend on its enzymatic activity.

- Unclear what the difference is between Figure 1B and Supplementary Figure 1B - how exactly is invasion being differentiated from migration given that a transwell system is being used for both?

Response: Thank you for this comment. Transwell inserts pre-coated **with** or without Matrigel (Corning, #354234) on the 8.0 μ M permeable polycarbonate membrane were used to determine **invasion** or migration potential of cells respectively, as described previously (PMID: 9428518, Cell, 1997). According to the official product manual, Matrigel is a soluble basement membrane extract of the Engelbreth-Holm-Swarm (EHS) mouse tumor that gels at room temperature to form a genuine reconstituted basement membrane. The major components of Corning Matrigel matrix are laminin (~60%), collagen IV (~30%), entactin (~8%) and heparan sulfate proteoglycan. When we conducted the transwell invasion assays, Matrigel was diluted in cold distilled water, added to the upper well of transwell chambers (20–30 μ g/well), and dried in a sterile hood. The Matrigel was then reconstituted with medium for one hour at 37°C before the addition of cells. For migration assays, the same procedure was followed, except that the transwell chambers were not coated with Matrigel. Reference and detailed methods have been

added to the **supplementary methods section of the appendix**, as quoted below:

“Matrigel invasion assays were performed as described (PMID: 9428518, Cell, 1997). Briefly, Matrigel was diluted in cold distilled water, added to the upper well of Transwell chambers (20–30 µg/well), and dried in a sterile hood. The Matrigel was then reconstituted with medium for one hour at 37°C before the addition of cells.”

- The metabolomic studies outlined in Figure 3 are interesting, but are not sufficient to conclude enhanced glycolysis. Changes in glycolytic intermediates can be independent of the flux through the pathway - they could increase with a block in glycolysis just as easily. To show enhanced glucose uptake or lactate synthesis, intracellular levels of glucose are also not sufficient. Instead, measurements on the media showing whether glucose is being depleted at a faster rate, and whether lactate is being exported at an increased rate will provide those data.

Response: Thanks very much for your valuable suggestion. We measured the levels of glucose and lactate in the media after culturing for 12 and 24 hr. Our results confirmed that PPA2 knockdown facilitated glucose consumption and lactate production in DLD1 cells under hypoxic stress (new Fig. 3F, G). Conversely, the opposite effect was observed with PPA2 overexpression under hypoxic stress (new Fig. EV3D, E). Furthermore, the knockout of HIF-1 α considerably weakened the increase in glucose consumption and lactate production induced by PPA2 knockdown in DLD1 cells (new Fig. EV3S, T). Detailed methods have been added to the **supplementary methods section of the appendix**. We appreciate your attention to this issue.

- In Figure 3H, CoCl₂ cannot be used as a hypoxia mimetic for metabolic experiments - high doses of cobalt ions can block other divalent ion transporters, and in doing so can alter metabolism. A small molecule PHD enzyme inhibitor (not the aKG-like DMOG) would be more suitable for this study.
- In general, it would be better if the authors could use a more recent, more specific PHD inhibitor in their studies, such as **IOX2**.

Response: We sincerely appreciate your suggestion, which has greatly contributed to the reliability of our study. We switched to use IXO2 as a hypoxia mimetic and repeated the extracellular acidification rate (ECAR) analysis, reaching the same conclusion as before, that PPA2 depletion, in combination with IOX2 treatment, significantly increased glycolysis and glycolytic capacity in DLD1 cells (new Fig. 3H-J). Conversely, overexpression of PPA2 exhibited an opposite effect on glycolysis in DLD1 cells (new Fig. EV3F-H). Additionally, we confirmed that, upon IOX2 stimulation, PPA2 knockdown promoted HIF-1 α protein level, while PPA2 overexpression inhibited it, consistent with the observations we made under hypoxic conditions (new Fig. EV4B).

- The data in Supplementary Figure 4B show that shPPA2 leads to increased

stabilization of HIF1a in the presence of CoCl₂. This does not necessarily rule out an activity for pVHL at this point - the proof for this would be to use VHL-deficient cells and overexpress PPA2 to show that this decreases its stability in this system. I believe that this experiment would significantly strengthen the manuscript

Response: We greatly appreciate your constructive suggestions and assistance in strengthening our study. Based on your suggestion, we overexpressed PPA2 in VHL depleted-DLD1 cells and performed immunoprecipitation and immunoblotting analyses. The results showed that: “the genetic inhibition of VHL stabilized HIF-1 α as expected, but did not abolish the effect of PPA2 on the protein and ubiquitination level of HIF-1 α (Page 11, new line 291-293, new Fig. EV4C, D).” Collectively, these results support a more reliable conclusion that PPA2 destabilizes HIF-1 α through ubiquitination and proteasomal degradation, independent of VHL. These findings align with the original aim of our research.

- From Line 341, the authors interrogate lysine modifications. It is a considerable leap to move straight to succinylation as the potential target modification, when there are a significant number of potential candidates, as tested in the blot in Figure 5A. Could the authors also refer to testing the other modifications shown?

Response: Thank you for this comment. In addition to the several acylation modifications displayed, we performed additional analysis of the lysine methylation and lactylation modifications of PPA2. The results indicate that: “PPA2 exhibited detectable levels of lysine acetylation and methylation, along with a lower level of lactylation, while malonylation was scarcely detected. No differences in these modifications were observed under hypoxic treatment. In contrast, PPA2 showed notable lysine succinylation modification under normoxic conditions, which gradually decreased with prolonged hypoxic treatment in both DLD1 and SW1116 cells (Page 13, new line 364-369, new Fig. 5A, B).” Therefore, we focus on the role of PPA2 succinylation in the hypoxic response of CRC tumor cells.

- Using a lysine to arginine point mutation to mimic non-succinylated lysine is not a common approach - could the authors reference this having been used before and demonstrated to work in this way. The same point is relevant for the use of a point mutation to glutamate to mimic succinylation - a reference is required demonstrating this point.

Response: Thank you for this comment. A point mutation replacing lysine with arginine or glutamate is used to mimic either the non-succinylated or succinylated state, as previously described in the studies quoted below:

“We also mutated K622 into arginine (K622R) or glutamate (K622E) to mimic its unmodified and succinylated states, respectively” (PMID: 38649537, *The EMBO Journal*, 2024).

“We constructed recombinant human ASS1 proteins with a lysine (K) to glutamic

acid (E) mutation at either K112 or K121 or at both sites to mimic succinylation on these sites” (PMID: 38448615, *Nature metabolism*, 2024).

“We next mutated GLS K311; K311-adjacent K245, K279, and K320; and K507 into Arg (R) and expressed these FLAG-tagged GLS mutants in 293T cells. Only GLS K311R significantly decreased succinyl-CoA-induced GLS succinylation.... These results indicated that GLS is succinylated at K311” (PMID: 33991485, *Molecular cell*, 2021).

“We generated a single succinylation mimetic (K-to-glutamine, E) mutant (LDHA^{K155E}) and succinylation-null (K-to-arginine, R) mutant (LDHA^{K155R}) and examined their enzymatic activity” (PMID: 37031273, *Cell death and differentiation*, 2023).

We have now added references in the results sections (Page 13-14, new line 377 and 394).

- The evidence for PPA2 K176 residue being involved in the desuccinylation in hypoxia is shown in the blot in Figure 5D. However, this level of evidence could mean that binding of something to this residue is required for desuccinylation of another residue, as much as it is the residue that is directly desuccinylated in hypoxia. The use of point mutants is not sufficient either, as this could also disrupt a putative binding site. Further direct proof is required to conclude that this residue is specifically desuccinylated in hypoxia

Response: Thank you for your valuable comment. To provide further direct proof that PPA2 K176 residue is directly desuccinylated under hypoxia, we developed specific antibody for PPA2 suK176 and performed additional experiments as quoted below:

“To further confirm this succinylated site in PPA2, we generated a specific antibody against suK176-PPA2. The specificity of this antibody was validated by its special recognition of suK176 in PPA2 WT but not in PPA2 K176R mutant (new Fig. 5E). Importantly, we observed that the level of suK176 decreased in a time-dependent manner under hypoxic stress, indicating that K176 is the critical site that is directly desuccinylated in hypoxic conditions (new Fig. 5E). (Page 14, new line 380-385).”

Furthermore, we detected increased PPA2 suK176 level following the treatment of the pan-sirtuin inhibitor nicotinamide (NAM) (new Fig. 5J) or siSIRT5 transfection (new Fig. 5K). In contrast, transfection with the SIRT5 WT plasmid, rather than the H158Y mutant, led to reduced PPA2 suK176 level (new Fig. 5N). Collectively, these results fully validated the specificity of the anti-suK176-PPA2 antibody and that hypoxia-induced PPA2 K176 desuccinylation was mediated by SIRT5. Detailed methods of generating the anti-suK176-PPA2 antibody were added to the **supplementary methods section of the appendix**.

- The data in Figure 6I is not sufficient to conclude that HIF1a protein and PPA2 protein are negatively correlated. The R2 values are very small, suggesting little correlation, despite a mathematical significant level being reached.

Response: Thank you for this comment. To further validate our conclusions, we increased the sample size to 90 cases and repeated the Pearson correlation analysis. We yielded results consistent with our previous findings and achieved a more convincing R^2 values (new Fig. 6I). We reviewed related research (as indicated below) and concluded that the new R^2 values were sufficient to establish a reliable negative correlation between PPA2 protein and HIF-1 α protein or its downstream targets.

PMID: 34077757, Molecular cell, 2021. “(I) Quantification of the immunochemical staining in... (n = 100).”

It should be noted that in the previous study, we used the R value to measure the strength and direction of the linear relationship between the two variables. Now, based on your comments, we have switched to using R^2 to better explain how well the linear regression model fits. We thank you for raising this issue and helping to enhance the credibility of our research.

Minor

- Lines 149-150; the evidence at this point is correlative, and nothing shows a direct role to here. I think the sentence should be changed to reflect that reduced expression of PPA2 is correlated with disease progression.

Response: We sincerely apologize for the incorrect statement and have revised the statement in the results section (Page 6, new line 151-152), as quoted below:

“These results strongly support that reduced expression of PPA2 is correlated with disease progression of CRC.”

- Similarly, the role that the authors are trying to elicit using their cell culture models is not PPA2 in progression, but in cell phenotype (line 163).

Response: Thank you for your careful review and valuable suggestions, and the statement has been revised in the results section (Page 6, new line 165-167), as quoted below:

“To explore and validate the function of PPA2 in cell phenotype, we designed two independent short hairpin RNAs (shRNAs) that target PPA2 (shPPA2) to stably knockdown its expression in DLD1 and SW1116 cells.”

- Methods for the metabolomics study does not reveal whether metabolite IDs were confirmed with chemical standards - can this be clarified in the methods section.

Response: Thank you for raising this question. We have added the statement to the **supplementary methods section of the appendix**, as quoted below:

“All metabolite standards were confirmed with chemical standards; their product information and chemical abstracts service (CAS) number were listed in Appendix Table 1.”

Appendix Table 1. Product information and CAS number of metabolite standards

Compounds	Product information	CAS
D-Glucose-6-phosphate (G6P)	#ZTR-G595338, TRC	56-73-5
D-Fructose-6-phosphate (F6P)	#ZG-23130, Glycosci	26177-86-9
Dihydroxyacetone-phosphate (DHAP)	#51269, Sigma	20915
Glyceraldehyde-3-phosphate (G3P)	#39705, Sigma	591-57-1
3-phosphoglycerate (3-PGA)	#P8877, Sigma	820-11-1
2-Phospho-D-glyceric acid (2-PGA)	#ZTR-P358000, TRC	3443-57-0
Phosphoenolpyruvic-acid (PEP)	#ZTR-P360870, Sigma	138-08-9
Pyruvic-acid (PA)	#RO10854, Sigma	138-08-9
Lactate	#L1750, Sigma	50-21-5
Cis-Aconitic-acid (CAA)	#ZC-66443, Zzstandard	585-84-2
Succinic-Acid (SA)	#ZL-158003, Zzstandard	2387-71-5
Fumaric-acid (FA)	#F110741, Aladdin	110-17-8
L-Glutamic-acid (Glu)	#ZC-45481, Zzstandard	56-86-0
Glutamine (Gln)	#CDAA-281494, ANPEL	56-85-9

Referee #2:

Summary:

The manuscript by Zhang, Di, and Wang et al., investigated the role of PPA2 and hypoxia in relation to colorectal cancer metastasis. The authors found that during periods of low oxygen PPA2 decreases HIF-1a stability via NEDD4. Furthermore, the authors showed that Sirt5 promotes dissociation of PPA2 and NEDD4 via succinylation and increases HIF-1a stability. Overall, the conclusions of this paper are justified by the data. I would recommend this manuscript for publication with some minor additions.

We appreciate your positive comments and insightful suggestions regarding our work. As recommended, we have incorporated relevant descriptions and conducted new experiments to enhance our study.

Our point-by-point responses are below:

Major Comments:

1. Can the authors show if the opposite metabolic alterations happen in PPA2 overexpression and sgHIF1a cells?

Response: Thank you for this comment. To address your concern, we knocked out HIF-1 α in PPA2-overexpressed DLD1 cells and performed additional metabolites detection. Results indicated that: "HIF-1 α knockout not only suppressed glucose consumption and lactate production as expected, but also abolished the effect of PPA2 depletion or overexpression on metabolites level (Page 10, new line 271-273, new Fig. 3S, T and EV3Q, R)."

2. How does OE of PPA2 influence downstream HIF target genes?

Response: We thank you for this comment. PPA2 overexpression inhibited the expression of downstream HIF target genes compared to the vector group in both CRC cell lines and hepatic injection liver metastatic models, as we originally illustrated in Figure EV3B, C.

3. Does PPA2 expression have an influence on primary tumor growth or only promotion of metastasis?

Response: Thank you for this comment. According to your suggestion, we have established a xenograft tumor model using DLD1 cells with either PPA2 knockdown or PPA2 overexpression. The results indicated that neither the knockdown nor the overexpression of PPA2 affected the growth of subcutaneous tumors (**as shown in Fig. A-F below**). The systematic screening strategy in Fig. 1A was designed to identify candidate genes associated with metastasis. The results of the bioinformatics analysis in Fig. 1B and immunohistochemical analyses in Fig. 1H, I further indicate that the

potential of PPA2 as a suppressive candidate gene in mCRC, given its characteristic low expression in stage IV (M1) tumors and liver metastatic foci. Therefore, in our study, we focus on the role of PPA2 in regulating tumor metastasis. We conclude that PPA2 influences metastasis but does not impact primary tumor growth.

Figure for reviewers removed

Subcutaneous injection of PPA2-knockdown (A-C) or PPA2-overexpression (D-E) DLD1 cells into BALB/c-nude mice to establish a xenograft model (n = 5). Tumor images (A, D), tumor volumes recorded at the indicated times (B, E) and tumor weights (C, F) are shown.

Data information: Data are mean \pm SD from five biological replicates (A-F). Statistics: two-way ANOVA (B, E); one-way ANOVA (C); unpaired two-tailed Student's t-test (F).

4. In general, does CRC actually experience hypoxia?

Response: Thank you for this comment. The most reliable measurement of tumor hypoxia is achieved using polarographic needle oxygen electrodes, commonly known as Eppendorf electrodes, which are inserted directly into tumors (PMID: 17536958, Antioxidants & redox signaling, 2007). Unfortunately, no studies have applied this method to assess the oxygenation status of CRC, likely due to the inherent challenges in accessing these tumors. As a result, all existing knowledge about CRC hypoxia has been derived from indirect methods. In 2013, Havelund *et al.* utilized positron emission tomography (PET) imaging to demonstrate a significantly higher uptake of the hypoxia-specific radiotracer 18F-fluoroazomycin-arabinofuranoside in rectal cancer (RC) compared to surrounding muscle and intestinal tissue, confirming the presence of hypoxia in RC (PMID: 23196674, Nuclear medicine communications, 2013). These findings are further supported by additional studies utilizing different hypoxic radiotracers (PMID: 18682881, Diseases of the colon and rectum, 2008) (PMID:18654902, Acta oncologica, 2008). Moreover, Wendling *et al.* demonstrated that the mean hemoglobin saturations were 42% in the tumor and 80% in healthy rectal wall biopsies, using a crayophotometric method (PMID: 6534106, Advances in experimental medicine and biology, 1984). Further evidence of hypoxia in CRC has

been gathered using pimonidazole, an exogenous hypoxia marker that selectively binds to intracellular macromolecules at low oxygen pressures. By analyzing several left colonic or rectal biopsies taken several hours after intravenous administration of pimonidazole, Goethals *et al.* found that, on average, 16.7% of the cells in CRC are in a hypoxic state (PMID: 16618579, International journal of radiation oncology, biology, physics, 2006). We have added the statement and references in the introduction section (Page 3, new line 80-82), as quoted below:

“Mounting studies have provided evidence for the existence of hypoxia in CRC (PMID: 16618579, International journal of radiation oncology, biology, physics, 2006) (PMID: 23196674, Nuclear medicine communications, 2013) (PMID: 6534106, Advances in experimental medicine and biology, 1984).”

Minor Comments:

1. 3R color legend order is different from how data presented in figure

Response: We appreciate you for raising this concern and we have corrected the color legend in the new Fig. 3R.

Dear Dr Wang,

Thank you for submitting your revised manuscript (EMBOJ-2024-119243R) to The EMBO Journal, as well for your patience with our response. Your amended study was sent back to the referees for their scientific re-evaluation, and we have received detailed comments from both of them, which I enclose below. As you will see, the experts state that the work has been substantially enhanced by the revisions and they are now broadly in favour of publication, pending minor revision.

Thus, we are pleased to inform you that your manuscript has been accepted in principle for publication in The EMBO Journal.

Please carefully consider the remaining minor points raised by reviewer #1 regarding data presentation, and adjust the related figure panels.

We also now need you to take care of a number of issues related to formatting and data presentation as detailed below, which should be addressed at re-submission.

Please contact me at any time if you have additional questions related to below points.

Thank you for giving us the chance to consider your manuscript for The EMBO Journal. I look forward to your final revision.

Again, please contact me at any time if you need any help or have further questions.

Kind regards,

Daniel Klimmeck

>> Author Contributions: Remove the author contributions information from the manuscript text. Note that CRediT has replaced the traditional author contributions section as of now because it offers a systematic machine-readable author contributions format that allows for more effective research assessment. and use the free text boxes beneath each contributing author's name to add specific details on the author's contribution.

More information is available in our guide to authors.
<https://www.embopress.org/page/journal/14602075/authorguide>

>> Figure callouts: Please ensure that the Table EV3 and Appendix Table S1 are called out correctly and in sequential order.; we don't use "supplementary" anymore so "supplementary information" on p20 should be updated.

>> Dataset EV legends: Tables EV3 and EV5 are the only ones that should stay EV tables as they are properly converted to PDF; the other three (Table EV1, EV2 and EV4) are more complex and should be updated to Dataset EV1-Dataset EV3 and the remaining two EV tables should then be Table EV1 and Table EV2 (the nomenclature should be updated in the source files, titles in our online system, and callouts throughout the manuscript).

>>Appendix file with ToC: needs to be a PDF file. "Supplementary" should be "Appendix"; the table needs to be renamed to Appendix Table S1. Please move all Supplementary Methods into the main manuscript Methods section.

>> Rename the current 'Materials and Methods' section to 'Methods'.

>> Remove the 'Supplementary documents' section from the manuscript.

>> Source data: source data should be uploaded as one (zipped) file per figure. All blots need to be presented as file per panel and need to be clearly labelled their respective folders, especially in folders 4 and 5; SD folders for EV figures could perhaps be combined into one zip folder.

>> Consider additional changes and comments from our production team as indicated below:

*Figure Legends

1. Please define the annotated p values ****/***/**/* as well as provide the exact p-values for the same in the legend of figure 1B, C, D, E, F, G, H, I; 2B, C, D, E, G, H, I, J, L, M, N; 3B, C, F, G, H, I, J, L, N, P, R, S, T, U; 5H, I; 6A, B, C, D, F, G; EV1 A-D; EV2 B, C, F, G, H; EV3 B, C, D, E, G, H, L, M, N, O, P, Q, R; EV5 F, G, H, I, J as appropriate.
2. Please note that the exact p values are not provided in the legends of figures 6I, EV3 A
3. Please indicate the statistical test used for data analysis in the legends of figures 3E, EV3 A
4. Please note that information related to n is missing in the legends of figures 1C

Referee #1:

The authors have addressed all my comments and I believe it is now of higher quality. One final small thing - the new data presented in Figure 2F and 2G (and associated EV panels), the glucose concentration change needs a unit for the y axis, and the lactate concentration change needs to be clear what the value is a percentage of (y axis again). Thanks

Referee #2:

The manuscript by Zhang, Di, and Wang et al., investigated the role of PPA2 and hypoxia in relation to colorectal cancer metastasis. The authors found that during periods of low oxygen PPA2 decreases HIF-1a stability via NEDD4. Furthermore, the authors showed that Sirt5 promotes dissociation of PPA2 and NEDD4 via succinylation and increases HIF-1a stability. The authors have responded satisfactorily to all my comments. After revision the conclusions of this paper are justified by the data. I would now recommend this manuscript for publication.

Responses letter

Thank you very much to the respected editor, all the staff and the two professional reviewers for recognizing our research and providing all the help. We have revised and provided the information according to your suggestions and comments. We are really delighted that our research can be officially published in The EMBO Journal. All the modifications made in the manuscript are highlighted in yellow. Our point-by-point responses are provided below.

>> Author Contributions: Remove the author contributions information from the manuscript text. Note that CRediT has replaced the traditional author contributions section as of now because it offers a systematic machine-readable author contributions format that allows for more effective research assessment. and use the free text boxes beneath each contributing author's name to add specific details on the author's contribution.

Response: We have removed the author contributions section from the manuscript text and identified the author's contribution on the submission system.

>> Figure callouts: Please ensure that the Table EV3 and Appendix Table S1 are called out correctly and in sequential order.; we don't use "supplementary" anymore so "supplementary information" on p20 should be updated.

Response: Thank you for your kind suggestion. We have ensured that all the tables are called out in the text in the correct order. It should be noted that the naming order of the tables has changed in the latest version.

>> we don't use "supplementary" anymore so "supplementary information" on p20 should be updated.

Response: We have revised the statement in the methods section (Page 26, line 751), as quoted below:

“Other data generated in this study are available in the article and appendix.”

>> Dataset EV legends: Tables EV3 and EV5 are the only ones that should stay EV tables as they are properly converted to PDF; the other three (Table EV1, EV2 and EV4) are more complex and should be updated to Dataset EV1-Dataset EV3 and the remaining two EV tables should then be Table EV1 and Table EV2 (the nomenclature should be updated in the source files, titles in our online system, and callouts throughout the manuscript).

>>Appendix file with ToC: needs to be a PDF file. "Supplementary" should be "Appendix"; the table needs to be renamed to Appendix Table S1. Please move all Supplementary Methods into the main manuscript Methods section.

>> Rename the current 'Materials and Methods' section to 'Methods'.
>> Remove the 'Supplementary documents' section from the manuscript.
>> Source data: source data should be uploaded as one (zipped) file per figure. All blots need to be presented as file per panel and need to be clearly labelled their respective folders, especially in folders 4 and 5; SD folders for EV figures could perhaps be combined into one zip folder.

Response: Thank you for your attention to these issues. The revision has been completed as you suggested.

>> Consider additional changes and comments from our production team as indicated below:

***Figure Legends**

1. Please define the annotated p values *****/***/**/* as well as provide the exact p-values for the same in the legend of figure 1B, C, D, E, F, G, H, I; 2B, C, D, E, G, H, I, J, L, M, N; 3B, C, F, G, H, I, J, L, N, P, R, S, T, U; 5H, I; 6A, B, C, D, F, G; EV1 A-D; EV2 B, C, F, G, H; EV3 B, C, D, E, G, H, L, M, N, O, P, Q, R; EV5 F, G, H, I, J as appropriate.

Response: Thanks for your comment. Instead of using *, we have directly indicated p-values in all the corresponding panels.

2. Please note that the exact p values are not provided in the legends of figures 6I, EV3A

Response: Thank you for your careful review. We have added p-values in Figure 6I and EV3A.

3. Please indicate the statistical test used for data analysis in the legends of figures 3E, EV3A

Response: Thank you for your thoughtful suggestion. We have specified the statistical test methods in the legend of Figure 3E and EV3A.

4. Please note that information related to n is missing in the legends of figures 1C

Response: Thank you for your careful review. We have directly marked the sample sizes of each group in the Figure 1C.

Reviewer comments:

Referee #1:

The authors have addressed all my comments and I believe it is now of higher quality. One final small thing - the new data presented in Figure 2F and 2G (and associated EV panels), the glucose concentration change needs a unit for the y axis, and the lactate concentration change needs to be clear what the value is a percentage of (y axis again). Thanks

Response: Thank you for your valuable suggestion. We believe that what you mentioned may refer to Figure 3F and 3G, and associated EV panels. Previously, we described glucose uptake ability using relative values, which is why the y-axis had no unit. Now, according to your suggestion, we have updated this to use the actual unit: $\mu\text{mol}/10^6$ cells, as illustrated in the revised Figure 3F, 3S, EV3D, EV3Q, and EV5G. Regarding the data processing of lactate concentration, we now consistently use the control group (shNT or Vector) at 24 hours as the normalization standard, as shown in the updated Figure 3G, 3T, EV3E, EV3R, and EV5H and corresponding legends.

Referee #2:

The manuscript by Zhang, Di, and Wang et al., investigated the role of PPA2 and hypoxia in relation to colorectal cancer metastasis. The authors found that during periods of low oxygen PPA2 decreases HIF-1a stability via NEDD4. Furthermore, the authors showed that Sirt5 promotes dissociation of PPA2 and NEDD4 via succinylation and increases HIF-1a stability. The authors have responded satisfactorily to all my comments. After revision the conclusions of this paper are justified by the data. I would now recommend this manuscript for publication.

Response: Thank you for your positive evaluation of our research and the valuable suggestions you put forward during the review process.

Dear Dr Wang,

Thank you for submitting the revised version of your manuscript. I have now evaluated your amended manuscript and concluded that the remaining minor concerns have been sufficiently addressed.

I am thus pleased to inform you that your manuscript has been accepted for publication in the EMBO Journal.

Related, I would like to hereby ask your consent on keeping the referee figures included in this file.

On a different note, I would like to alert you that EMBO Press offers a format for a video-synopsis of work published with us, which essentially is a short, author-generated film explaining the core findings in hand drawings, and, as we believe, can be very useful to increase visibility of the work. Please see the following link for representative examples and their integration into the article web page:

<https://www.embopress.org/doi/full/10.15252/emj.2019103932>

Best regards,

Daniel Klimmeck

Daniel Klimmeck, PhD
Senior Editor
The EMBO Journal
EMBO
Postfach 1022-40
Meyerohofstrasse 1
D-69117 Heidelberg
contact@embojournal.org